# Development of a deep neural network for predicting 6-hour average PM$_{2.5}$ concentrations up to two subsequent days using various training data

Jeong-Beom Lee[1], Jae-Bum Lee[1], Youn-Seo Koo[2], Hee-Yong Kwon[3], Min-Hyeok Choi[1], Hyun-Ju Park[1], Dae-Gyun Lee[1]

[1]Air Quality Forecasting Center, National Institute of Environmental Research (NIER), Incheon, 22689, South Korea
[2]Department of Environmental and Energy Engineering, Anyang University, Gyeonggi, 14028, South Korea
[3]Department of Computer Engineering, Anyang University, Gyeonggi, 14028, South Korea

*Correspondence to*: Jae-Bum Lee (gercljb@korea.kr)

**Abstract.** This study aims to develop a deep neural network (DNN) model as an artificial neural network (ANN) for the prediction of 6-hour average fine particulate matter (PM$_{2.5}$) concentrations for a three-day period—the day of prediction (D+0), one day after prediction (D+1) and two days after prediction (D+2)—using observation data and forecast data obtained via numerical models. The performance of the DNN model was comparatively evaluated against that of the currently operational Community Multiscale Air Quality (CMAQ) modelling system for air quality forecasting in South Korea. In addition, the effect on predictive performance of the DNN model on using different training data was analyzed. For the D+0 forecast, the DNN model performance was superior to that of the CMAQ model, and there was no significant dependence on the training data. For the D+1 and D+2 forecasts, the DNN model that used the observation and forecast data (DNN-ALL) outperformed the CMAQ model. The root-mean-squared error (RMSE) of DNN-ALL was lower than that of the CMAQ model by 2.2 μgm$^{-3}$, and 3.0 μgm$^{-3}$ for the D+1 and D+2 forecasts, respectively, because the overprediction of higher concentrations was curtailed. An Index Of Agreement (IOA) increase of 0.46 for D+1 prediction and 0.59 for the D+2 prediction was observed in case of the DNN-ALL model compared to the IOA of the DNN model that used only observation data (DNN-OBS). In additionally, An RMSE decrease of 7.2 μgm$^{-3}$ for the D+1 prediction and 6.3 μgm$^{-3}$ for the D+2 prediction was observed in case of the DNN-ALL model, compared to the RMSE of DNN-OBS, indicating that the inclusion of forecast data in the training data greatly affected the DNN model performance. Considering the prediction of the 6-hour average PM$_{2.5}$ concentration, the 8.8 μgm$^{-3}$ RMSE of the DNN-ALL model was 2.7 μgm$^{-3}$ lower than that of the CMAQ model, indicating the superior prediction performance of the former. These results suggest that the DNN model could be utilized as a better-performing air quality forecasting model than the CMAQ, and that observation data plays an important role in determining the prediction performance of the DNN model for D+0 forecasting, while prediction data does the same for D+1 and D+2 forecasting. The use of the proposed DNN model as a forecasting model may result in a reduction in the economic losses caused by pollution-mitigation policies and aid better protection of public health.

# 1 Introduction

Fine particulate matter ($PM_{2.5}$) refers to tiny particles or droplets in the atmosphere that exhibit an aerodynamic diameter of less than 2.5 μm. Such matter is mainly produced through secondary chemical reactions following the emission of precursors, such as sulfur oxides ($SO_X$), nitrogen oxides ($NO_X$), and ammonia ($NH_3$), into the atmosphere (Kim et al., 2017). Studies reveal that the $PM_{2.5}$ generated in the atmosphere is introduced into the human body through respiration and increases the incidence of cardiovascular and respiratory diseases as well as premature mortality (Pope et al., 2019; Crouse et al., 2015). To reduce the negative effects on health caused by $PM_{2.5}$, the National Institute of Environmental Research (NIER) under the Ministry of Environment of Korea has been performing daily average $PM_{2.5}$ forecasts for 19 regions since 2016. The forecasts rely on the judgment of the forecaster based on the Community Multiscale Air Quality (CMAQ) prediction results and observation data. The forecasts are announced four times daily (at 05:00, 11:00, 17:00, and 23:00 (LST)), and the predicted daily average $PM_{2.5}$ concentrations are represented via four different air quality index (AQI) categories in South Korea: good ($PM_{2.5} \leq 15$ μgm$^{-3}$), moderate (16 μgm$^{-3} \leq PM_{2.5} \leq 35$ μgm$^{-3}$), bad (36 μgm$^{-3} \leq PM_{2.5} \leq 75$ μgm$^{-3}$), and very bad (76 μgm$^{-3} \leq PM_{2.5}$). When the forecasts were based on the CMAQ model, the accuracy (ACC) of the daily forecast for the following day (D+1) in Seoul, South Korea, over the three-year period from 2018 to 2020 was 64%, and the prediction accuracy for the high-concentration categories ("bad" and "very bad") was 69%. Furthermore, a high false-alarm rate (FAR) of 49% was obtained. Studies have revealed that the prediction performance of the atmospheric chemical transport model (CTM) is limited by the uncertainties in the meteorological field data used as model input (Seaman, 2000; Doraiswamy et al., 2010; Hu et al., 2010; Jo et al., 2017; Wang et al., 2021), and in emissions (Hanna et al., 2001; Kim and Jang, 2014; Hsu et al., 2019). Moreover, the physical and chemical mechanisms in the model cannot fully reflect real-world phenomena (Berge et al., 2001; Liu et al., 2001; Mallet and Sportisse, 2006; Tang et al., 2009).

To overcome the uncertainty and limitations of the atmospheric CTM, a model for predicting air quality using artificial neural networks (ANNs) based on statistical data has recently been developed (Cabaneros et al., 2019; Ditsuhi et al., 2020). Studies using ANNs, such as the recurrent neural network (RNN) algorithm that is advantageous for time-series data training (Biancofiore et al., 2017; Kim et al., 2019; Zhang et al., 2020; Huang et al., 2021) and deep neural network (DNN) algorithm that is advantageous for extracting complex and non-linear features, are underway (Schmidhuber et al., 2015; LeCun et al., 2015; Lightstone et al., 2017; Cho et al., 2019; Eslami et al., 2020; Chen et al., 2021; Lightstone et al., 2021). Kim et al. (2019) developed an RNN model to predict $PM_{2.5}$ concentrations after 24-hour periods at two observation points in Seoul. The evaluation of the prediction performance of the RNN model for the May–June, 2016 period yielded an index of agreement (IOA) range between 0.62 and 0.76, which constituted a 0.12 to 0.25 IOA improvement compared to the CMAQ model. Lightstone et al. (2021) developed a DNN model that predicted 24-hour $PM_{2.5}$ concentrations based on aerosol optical depth (AOD) data and Kriging $PM_{2.5}$. The DNN-model predictions for the January–December, 2016 period yielded a root-mean-squared error (RMSE) of 2.67 μgm$^{-3}$, thereby demonstrating a prediction-performance improvement of 2.1 μgm$^{-3}$ compared to the CMAQ model.

It is to be noted that previous studies concerning the prediction of $PM_{2.5}$ concentrations using ANNs primarily developed and evaluated models for predicting the daily average concentration within a 24-hour period based solely on observation data. In this study, we developed a DNN model that predicts $PM_{2.5}$ concentrations at 6-hour intervals over three days—from the day of prediction (D+0) to two days after the day of prediction (D+2)—by extending the prediction period compared to the previous studies. Furthermore, the daily and 6-hour average prediction performance was comparatively evaluated against that of the CMAQ model currently operational for such predictions. In addition, the effect of the training data on the daily prediction performance of the DNN model was quantitatively analyzed via three experiments that used different configurations of the training data in terms of predictive data from numerical models as well as observation data.

## 2 DNN model implementation and acquisition of training data

Figure 1 outlines the process for the development of the DNN model used herein, which consists of three broad stages: preprocessing, model training and post-processing. In the preprocessing stage, the data necessary for the development of the DNN model are collected, and the collected data are processed into a suitable format for use as the training and validation data. In the model training stage, the backpropagation algorithm and parameters are applied to implement the DNN model, and the most optimal "weight file" is saved once training and validation are complete. In the post-processing stage, prediction is performed using the saved "weight file". Sect. 2.1 provides a detailed description of the data used for training, and Sect. 2.2 describes the development of the DNN model.

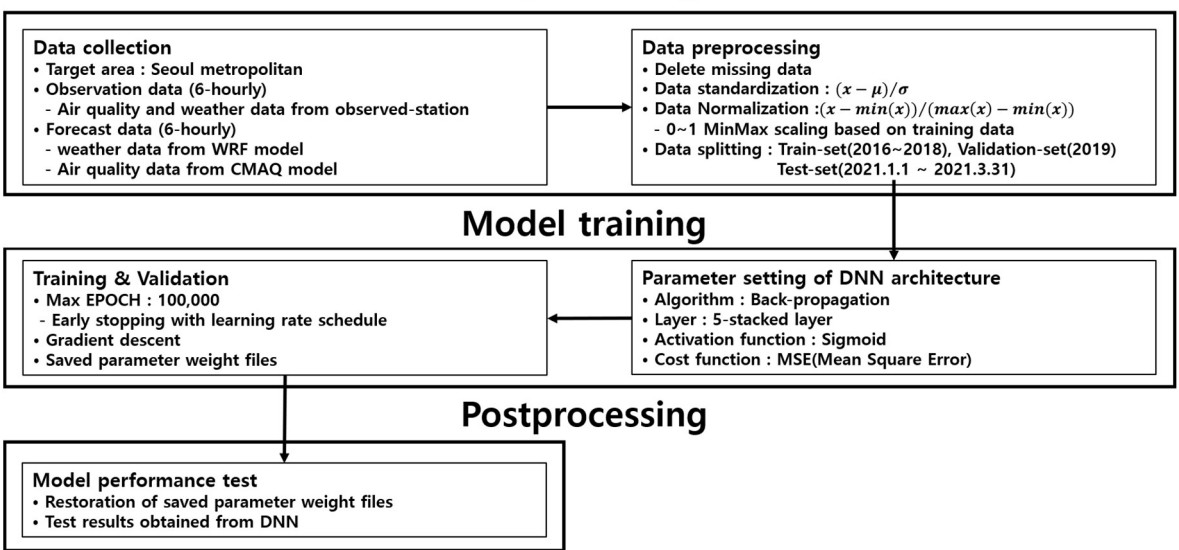

**Figure 1. Flowchart of the PM$_{2.5}$ forecasting system based on the DNN algorithm.**

## 2.1 Training data acquisition

For training of the DNN model, validating the trained DNN model, and making predictions using the developed DNN model, we used observation data, such as ground-based air quality and weather data, as well as forecasting data, such as ground-based and altitude-specific weather data and ground-based PM$_{2.5}$, generated via the WRF and CMAQ models in Seoul, South Korea. In addition, the membership function was used to reflect temporal information. Data pertaining to a three-year period (2016 to 2018) were used for training the model, and data pertaining to 2019 were used for validation. Data pertaining to a three-month period (January–March 2021) were used to evaluate the prediction performance.

Figure 2 illustrates the spatial distribution of the weather and air quality observation points in Seoul, South Korea, where the observation data used for training the model had been measured, and Table 1 presents a list of the weather and air quality observation data variables used for the training. Six variables of air quality (SO$_2$, NO$_2$, O$_3$, CO, PM$_{10}$, and PM$_{2.5}$), measured with the measuring equipment provided by Air Korea on their website, were used to obtain observation data. SO$_2$ and NO$_2$ are the precursors that directly affect the changes in the PM$_{2.5}$ concentration. O$_3$ is generated by NOx and volatile organic compounds (VOCs) and causes direct and indirect effects on the changes in the PM$_{2.5}$ concentration (Wu et al., 2017; Geng et al., 2019). CO affects the generation of O$_3$ in the oxidation process via the OH reaction, which, in turn, has

an indirect effect on the changes in the PM$_{2.5}$ concentration (Kim et al., 2016). Furthermore, particulate matter with particles exhibiting a less than 10 μm diameter (PM$_{10}$) is highly correlated with PM$_{2.5}$ during periods of high concentration and exhibits similar trends in seasonal concentrations

(Mohammed et al., 2017; Gao and Ji, 2018).

Real-time data from the Automated Surface Observing System (ASOS) were used as the weather data, through the uniform resource locator-application programming interface (URL-API) operated by the Korea Meteorological Administration. The eight variables for the surface-weather data included: vertical and horizontal wind speed, precipitation, relative humidity, dew point, atmospheric pressure, solar radiation, and temperature. Wind speeds and precipitation are known to be negatively correlated with the PM$_{2.5}$ concentration, whereas an

increase in the relative humidity increases the PM$_{2.5}$ concentration. Wind speed is generally associated with turbulence, and an increase in the intensity of the turbulence facilitates the mixing of air, inducing a decrease in the PM$_{2.5}$ concentration (Yoo et al., 2020). Precipitation affects the PM$_{2.5}$ concentration owing to the washing effect therein. A lower than 80% increase in the relative humidity affects the increase in the PM$_{2.5}$ concentration, owing to increased condensation and nucleation (Yoo et al., 2020; Kim et al., 2020). The dew point is associated with relative humidity; therefore, it has an indirect effect on the PM$_{2.5}$ concentration. In addition, atmospheric pressure, solar

radiation, and temperature affect the occurrence of high PM$_{2.5}$ concentrations and seasonal changes in PM$_{2.5}$. In terms of atmospheric pressure, the atmospheric stagnation caused by high pressure influences the occurrence of high PM$_{2.5}$ concentrations (Park and Yu, 2018). Solar radiation appears to be negative correlated with the PM$_{2.5}$ concentration in winter (Turnock et al., 2015), and temperature is reported to affect the changes in the PM$_{2.5}$ concentration owing to an increased sulfate concentration and decreased nitrate concentration at high temperatures (Dawson et al., 2007; Jacob and Winner, 2009).

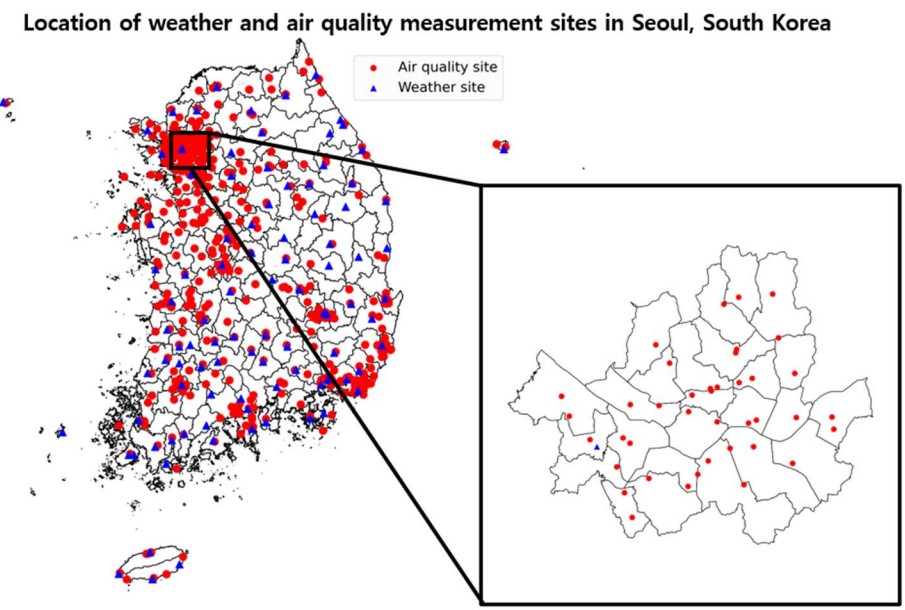

**Figure 2. Spatial distributions of weather (▲) and air quality (●) measurement sites in Seoul.**

Table 1. Training variables in the PM$_{2.5}$ prediction system using a DNN based on surface-weather observations. Air quality variables were obtained from 41 air quality measurement equipment in Seoul. Surface weather variables were obtained from ASOS in Seoul. Observation

data were collected every hour.

| Observation Variable | Description | Unit |
|---|---|---|
| O_SO$_2$ | Sulfur dioxide | ppm |
| O_NO$_2$ | Nitrogen dioxide | ppm |
| O_O$_3$ | Ozone | ppm |
| O_CO | Carbon monoxide | ppm |
| O_PM$_{10}$ | Particulate matter (aerodynamic diameters $\leq$ 10 μm) | μgm$^{-3}$ |
| O_PM$_{2.5}$ | Particulate matter (aerodynamic diameters $\leq$ 2.5 μm) | μgm$^{-3}$ |
| O_V | Vertical wind velocity | m/s |
| O_U | Horizontal wind velocity | m/s |
| O_RN_ACC | Accumulative precipitation | Mm |
| O_RH | Relative humidity | % |
| O_Td | Dew point temperature | °C |
| O_Pa | Pressure | hPa |
| O_Radiation | Solar radiation | 0.01 MJ per hr-m$^3$ |
| O_Ta | Air temperature | °C |


Figure 3 depicts the nested-grid modeling domains used to generate the forecast data in terms of surface-level and altitudinal weather and air quality that is used for training the DNN model, with Northeastern Asia represented as Domain 1 (27 km), and the Korean Peninsula represented as Domain 2 (9 km). The simulation results of the Weather Research and Forecasting (WRF, v3.3) model, a regional-scale weather model developed by the National Center for Environmental Prediction (NCEP) under the National Oceanic and Atmospheric

Administration (NOAA) in the United States, were used as the weather forecast data. The simulation results obtained via the CMAQ system (v4.7.1) developed by the US EPA were used as the PM$_{2.5}$ prediction data. The Unified Model (UM) global forecast data provided by the Korea Meteorological Administration were used as the initial and boundary conditions of the WRF model for the weather simulation. In the WRF model simulation, the Yonsei University Scheme (YUS) (Hong et al., 2006) was used for the planetary boundary layer (PBL) physics, the WRF single-moment class-3 (WSM3) scheme (Hong et al., 1998; Hong et al., 2004) was used for cloud

microphysics, and the Kain–Fritsch scheme (Kain, 2004) was used for cloud parameterization. The meteorological field generated was converted into a form of data input to the air quality numerical model using the Meteorology-Chemistry Interface Processor (MCIP, v3.6). The Sparse Matrix Operator Kernel Emission (SMOKE, v3.1) model was applied to the emissions inventory of Northeastern Asia (excluding South Korea). The Model Inter-Comparison Study for Asia, Phase 2010 (MICS-Asia; Itahashi et al., 2020) and the Clean Air Policy Support System, 2010 (CAPSS) were applied to the emissions inventory of South Korea. The Model of Emissions of Gases and

Aerosols from Nature (MEGAN, v2.0.4) was used to represent natural emissions. In case of the CMAQ model for PM$_{2.5}$ concentration simulation, the Statewide Air Pollution Research Center, version 99 (SAPRC-99; Carter et al., 1999) mechanism was used for the chemical mechanism, the fifth-generation CMAQ aerosol module (AERO5; Binkowski et al., 2003) was used for the aerosol mechanism, and the

Yamartino scheme for mass-conserving advection (YAMO scheme) (Yamartino, 1993) was used for the advection process. We directly generated the training data using the WRF and CMAQ.

Table 2 presents a list of the weather and air quality prediction model data variables used for training the $PM_{2.5}$ prediction system. The air quality forecast variable of the CMAQ model was $PM_{2.5}$. 16 meteorological forecast variables were created by the WRF model. $PM_{2.5}$ and its precursors are emitted from the ground, and they move at an altitude of 1.5 km or less. Therefore, lower altitude data variables were mainly used. The meteorological forecast variables on the ground included vertical and horizontal wind speed, precipitation, relative humidity, atmospheric pressure, temperature, and mixing height. In addition, the predicted meteorological variables for each altitude

included the geopotential height as well as the vertical and horizontal wind speed at 925 hPa. The geopotential height, vertical and horizontal wind speed, relative humidity, potential temperature at 850 hPa, and the difference in the potential temperature between 850 and 925 hPa were also included. An increase or decrease in mixing height, which depends on thermal and mechanical turbulence, affects the spread of air pollutants. As the mixing height increases, the diffusion intensity increases and the concentration of air pollutants, such as $PM_{2.5}$, decreases. The potential temperature is an indicator of the vertical stability of the atmosphere, and the vertical stability can be used

to identify the formation of the inversion layer, which has a significant effect on the $PM_{2.5}$ concentration (Wang et al., 2014). Finally, altitude data are associated with the atmospheric stability and long-term transport of air pollutants (Lee et al., 2018).

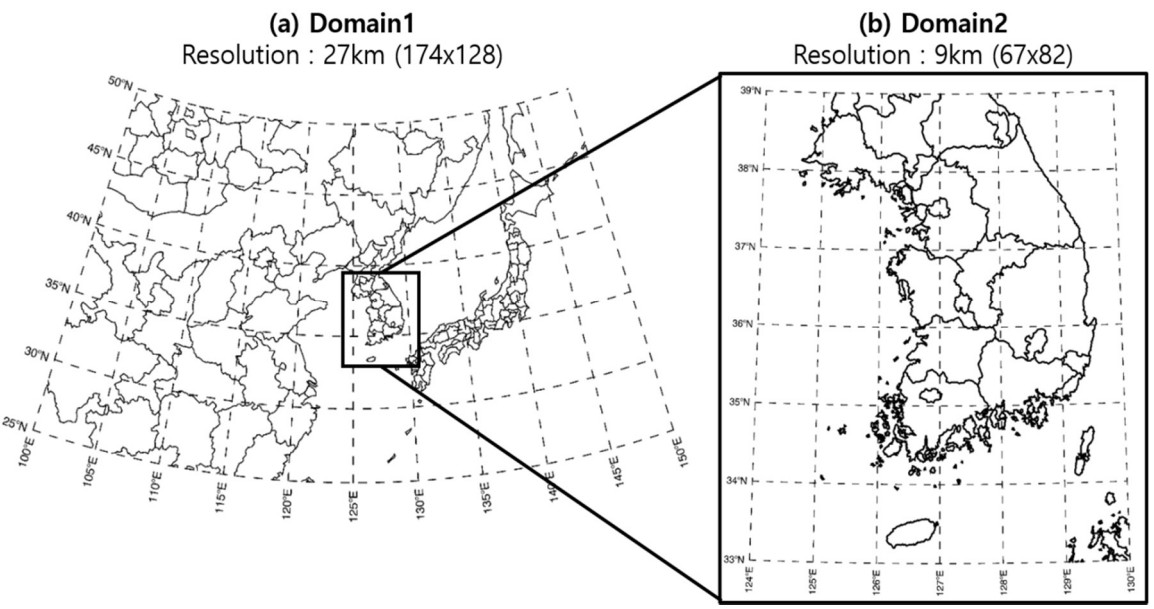

**Figure 3. CMAQ modeling domains applied to generate the DNN model training data: (a) Northeast-Asian area with 27 km horizontal grid resolution and (b) Korean-Peninsula area with 9 km horizontal grid resolution.**

Table 2. Training variables in the $PM_{2.5}$ prediction system using a DNN based on the WRF and CMAQ models. WRF and CMAQ model results were obtained from 9 km horizontal grid resolution. These values are collected on an hourly interval.

| Model | Forecast Variable | Description | Unit |
|---|---|---|---|
| CMAQ | F_$PM_{2.5}$ | Particulate matter (aerodynamic diameter $\leq$ 2.5 μm) | μgm$^{-3}$ |

| | | |
|---|---|---|
| F_V | Vertical wind velocity at surface | m/s |
| F_U | Horizontal wind velocity at surface | m/s |
| F_RN_ACC | Accumulative precipitation | Mm |
| F_RH | Relative humidity at surface | % |
| F_Pa | Pressure at surface | Pa |
| F_Ta | Air temperature at surface | K |
| F_MH | Mixing height | M |
| F_925hpa_gpm | Position altitude at 925 hPa | M |
| F_925hpa_V | Vertical wind velocity at 925 hPa | m/s |
| F_925hpa_U | Horizontal wind velocity at 925 hPa | m/s |
| F_850hpa_gpm | Position altitude at 850 hPa | M |
| F_850hpa_V | Vertical wind velocity at 850 hPa | m/s |
| F_850hpa_U | Horizontal wind velocity at 850 hPa | m/s |
| F_850hpa_RH | Relative humidity at 850 hPa | % |
| F_850hpa_Ta | Potential temperature at 850 hPa | Θ |
| F_Temp_ 850hpa-925hpa | Potential temperature difference between 850 hPa and 925 hPa | Θ |

To train the DNN model to understand the change-patterns in the $PM_{2.5}$ concentration over time and consider the propagation of temporal change, time data were generated using the membership function presented by Yu et al. (2019). The concept of the membership function is derived from the fuzzy theory, and it defines the probability that a single element belongs to a set. In this study, the probability that the date (element) belongs to 12 months (set) was calculated using the membership function. $PM_{2.5}$ concentration in Seoul is high in January, February, March, and December, and low from August to October. $PM_{2.5}$ concentration has a characteristic that changes gradually from month to month. The membership function was used to reflect these monthly change characteristics. The temporal data using the membership function contained twelve variables, representing the months from January to December. The sum of the variables was set to 1. Of the twelve variables, ten had a value of 0, and two had values between 0 and 1. The two non-zero variables were determined based on the day of generation of the temporal data and were defined as "month" and "adjacent month". If the temporal data were generated between the first to the fourteenth day of a "month", the "adjacent month" referred to the month preceding this "month". If the temporal data were generated between the sixteenth to the thirty-first day of a "month", the "adjacent month" referred to the month succeeding this "month". The "adjacent month" was not considered when the temporal data were generated on the fifteenth day of the "month". The values of the "adjacent month" and "month" variables were calculated through Eq. (1) to Eq. (4). For example, when generating the temporal data for January 10, the "month" would be January, and the "adjacent month" would be December. Based on the calculations in Eq. (1), the "month" variable value would equal 0.82 and the "adjacent month" variable value would equal 0.18, and the rest of the variable values from February to November would equal 0.

$$\text{If (Day} < 15) \text{ then } '\text{Month value}' = \frac{1}{28} \times \text{Day} + \frac{13}{28}, \qquad (1)$$

If (Day > 15) then $'Month\ value' = -\frac{1}{30} \times Day + \frac{3}{2}$,         (2)

If (Day = 15) then $'Month\ value' = 1$,         (3)

$'Adjacent\ Month\ value' = 1 - 'Month\ value'$,         (4)

## 2.2 Implementation of the DNN model

To develop DNN models over six-hour intervals, time steps (T-steps) were constructed for the target period of three days (D+0 to D+2) to
perform predictions as shown in Table 3. $T_{12\_D0}$ to $T_{24\_D0}$ are included in the day of prediction (D+0), $T_{06\_D1}$ to $T_{24\_D1}$ in the one day after of prediction (D+1), $T_{06\_D2}$ to $T_{24\_D2}$ in the two days after of prediction (D+2). Weather and air quality prediction data used in each T-step training data averages one-hour interval data into six-hour interval data. And the 9km grids corresponding Seoul were averaged, spatially. The observation data used in each T-step training data averages the preceding six-hour period at the beginning of the forecast (01:00 to 06:00 on D+0).


Table 3. Configuration of the training data for each T-step to implement the DNN model for the 6-hour average prediction

| Day | T-step | Time | Configuration of the training data |
|-----|--------|------|-----------------------------------|
| D+0 | $T_{12\_D0}$ | 07:00 to 12:00 | |
| | $T_{18\_D0}$ | 13:00 to 18:00 | |
| | $T_{24\_D0}$ | 19:00 to 00:00 | |
| D+1 | $T_{06\_D1}$ | 01:00 to 06:00 | |
| | $T_{12\_D1}$ | 07:00 to 12:00 | 01:00 to 06:00 observations data on D+0 at each T-step |
| | $T_{18\_D1}$ | 13:00 to 18:00 | + |
| | $T_{24\_D1}$ | 19:00 to 00:00 | Forecast data of $T_{x\_Dy}$ (x: 06, 12, 18, 24, y: 0 to 2) from CMAQ and WRF |
| D+2 | $T_{06\_D2}$ | 01:00 to 06:00 | |
| | $T_{12\_D2}$ | 07:00 to 12:00 | |
| | $T_{18\_D2}$ | 13:00 to 18:00 | |
| | $T_{24\_D2}$ | 19:00 to 00:00 | |

The feature scaling, including standardization and normalization, was implemented to transform data into uniform formats, reduce data bias of training data, and ensure equal training for the DNN model at each T-step. The normal distribution of the variables in the training data was standardized through standardization. The variables in the training data were standardized to be distributed in the range of a mean
of 0 and standard deviation of 1. The standardized variables of the training data were subsequently normalized to the minimum (min(x)) and maximum (max(x)) values so that the values would be bounded in an equal range between 0 and 1. Both normalization and standardization were applied to train the characteristics of training variables equally to the DNN model. Standardization and normalization were performed using the *Z*-score (Eq. (5)) and Min-max scaler (Eq. (6)), respectively.

$Z\text{-score} = \frac{x - \mu}{\sigma}$,         (5)

$\text{Min-max scaler} = \frac{x - \min(x)}{\max(x) - \min(x)}$,         (6)

Figure 4 depicts the training process of the DNN model. After feature scaling, the training data is trained through the backpropagation algorithm in the five-stacked-layer DNN model. The statistical and AQI performance results of the DNN model based on the layer are presented in Table S1 and S2 in the Supplement, respectively. The results of the four-stacked-layer and five-stacked-layer models show that the performance is similar. However, compared with the four-stacked-layer model, the RMSE of the five-stacked-layer decreases by approximately 0.1 $\mu$gm$^{-3}$ to 1 $\mu$gm$^{-3}$ at D+0 to D+2, and the ACC of the five-stacked-layer model increases by approximately 1 %p to 6 %p at D+0 to D+2. Therefore, the five-stacked-layer model shows the best performance. The six-stacked-layer and eight-stacked-layer models contain errors that converge without decreasing during the training process of the model (vanishing gradient problem). The cause of this problem is the activate function. The backpropagation algorithm consists of the feedforward and backpropagation processes. Feedforward is the process of calculating the difference (cost) between the output value (hypothesis) and target value (true value) in the output layer, after the calculation has proceeded from the input layer to subsequent layers and finally reached the output layer. Backpropagation is the process of creating new node values for the input layer by updating the weight using the cost calculated in the feedforward process.

In the feedforward process, the node (i) value ($x_i^{(l)}$) of the previous layer (l) is converted to the hypothesized ($x_i^{(l+1)}$), the node (m) value of the subsequent layer ($l + 1$) is converted through the weight ($w_{m,i}^l$), deviation ($b_m$), and sigmoid function ($\emptyset(Z_m^{(l+1)})$), which is an activation function. Equations (7) and (8) outline the calculation process.

$$\mathbf{Z_m^{(l+1)}} = \sum_{i=1}^{n}(\mathbf{x_i^{(l)}} \times \mathbf{w_{m,i}^{(l)}} + \mathbf{b_m}) \, , \tag{7}$$

$$\mathbf{x_m^{(l+1)}} = \emptyset\left(\mathbf{Z_m^{(l+1)}}\right) = \frac{1}{1+e^{(-Z_m^{(l+1)})}} \, , \tag{8}$$

The mean squared error (MSE), a cost function, is applied to the difference (cost) between the hypothesized and target value calculated during the forward propagation process, as denoted by Eq. (9) (Hinton and Salakhutdinov, 2006).

$$\text{Cost} = \frac{1}{n}(\mathbf{x_m^{\text{Outlayer}}} - \text{Target})^2 = \frac{1}{n}(\text{Hypothesis} - \text{Target})^2 \, , \tag{9}$$

In the backpropagation process, the weights calculated in the feedforward process are updated via the gradient descent method. For weight updating, the corresponding magnitude can be adjusted by multiplying it with a scalar value known as the learning rate ($\eta$) (Eq. (10)) (Bridle, 1990).

$$\mathbf{W_{m,i}^{(l)}} = \mathbf{W_{m,i}^{(l)}} - \eta \frac{\partial \text{Cost}}{\partial \mathbf{w_{m,i}^{(l)}}} \, , \tag{10}$$

Therefore, the backpropagation algorithm is configured as expressed in Eq. (5) to Eq. (10), and the DNN model learns the features of the training data by repeating the backpropagation algorithm as many times as the number of epochs.

In this study, early-stopping was applied to avoid the overfitting that occurred in the form of a decrease in the cost of the training data while the cost of the validation data increased with the number of epochs. The early-stopping condition is applicable when the cost value of the validation data at $\text{Epoch}_n$ is lower than the cost of the validation data from $\text{Epoch}_{n+1}$ to $\text{Epoch}_{Max}$. When the early-stopping condition is satisfied, the user-defined variable "Count" increases by 1 if the "Count" is zero, and if "Count" is non-zero, the learning rate decreases by $10^{-1 \times Count}$, so that learning is performed with an updated learning rate from $\text{Epoch}_{n+1}$ onwards. When the cost values of the validation data from $\text{Epoch}_{n+1}$ to $\text{Epoch}_{Max}$ exceed the cost values of $\text{Epoch}_n$ in the previous "Count," the learning of the model is completed.

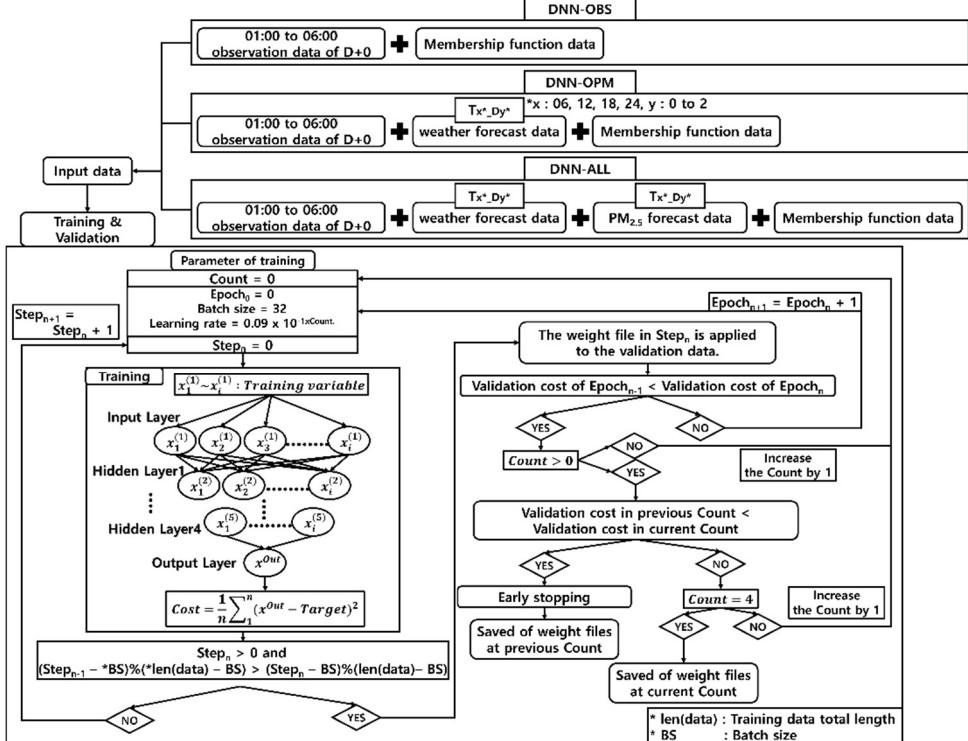

220 **Figure 4. Structure of DNN model training process.**

## 3 Experimental design and indicators for prediction performance evaluation

Figure 5 displays the average monthly PM$_{2.5}$ concentrations observed in Seoul from 2016 to 2019. The highest average monthly PM$_{2.5}$ concentration between 2017 and 2019 was observed in January, March, and December, i.e., during the winter season. The average monthly PM$_{2.5}$ concentration ranged between 28.8 and 37.8 μgm$^{-3}$ in winter and 16.6 and 26.6 μgm$^{-3}$ in summer over the four-year period (2016 to 225 2019). This indicated that the concentration in winter exceeded that in summer by approximately 12 μgm$^{-3}$. In this study, the prediction performance of the DNN model was evaluated during winter months (January 1, 2021, to March 31, 2021) that exhibited high PM$_{2.5}$ concentrations.

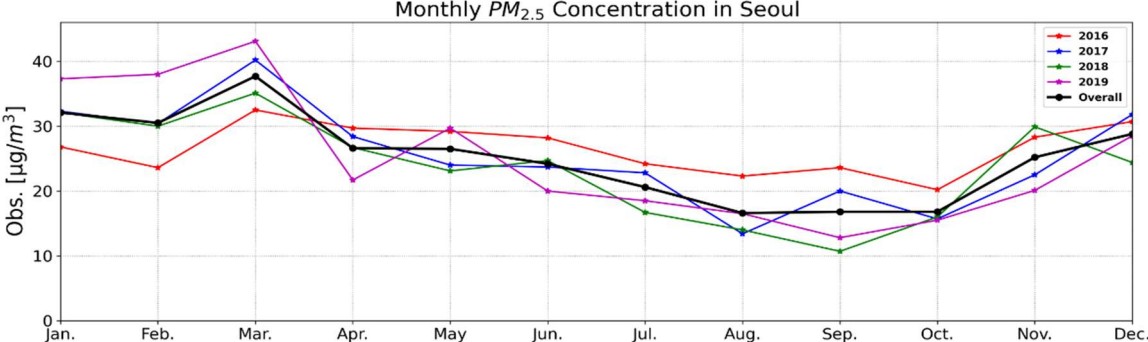

**Figure 5. Time series of the monthly average PM2.5 concentrations from 2016 to 2019.**

230 Three experiments (DNN-OBS, DNN-OPM, and DNN-ALL) were performed to examine the effects of the training-data configuration on the prediction performance of the DNN model. The DNN-OBS model used the observation data as the sole training data, the DNN-OPM model used both observation and weather forecast data for $T_{x\_Dy}$ (x: 06, 12, 18, 24, y: 0 to 2) as the training data, and the DNN-ALL model used the observation data, weather forecast data, and PM2.5 concentration prediction data $T_{x\_Dy}$ (x: 06, 12, 18, 24, y: 0 to 2) as the training data. The observation variables presented in Table 1 in Sect. 2.1 were used as common variables in the three experiments. Among the 235 predictors shown in Table 2 in Sect. 2.1, the variables produced in the WRF model were used in the DNN-OPM and DNN-ALL models, whereas the variables produced in the CMAQ model were used only in the DNN-ALL model.

The prediction performances of the three DNN-model experiments were evaluated based on statistics and the AQI. The MSE, RMSE, IOA, and correlation coefficient (R) were used as the indicators in statistical evaluation. The MSE and RMSE, which represented the loss functions of the DNN model, were used to determine the quantitative difference between the model predictions and observed values. The 240 IOA indicator determined the level of agreement between the model predictions and observed values based on the ratio of the MSE to the potential error. The R indicator determined the correlation between the model predictions and observed values. Equations (11) to (14) were used to calculate these five indicators.

$$\text{MSE } (\mu gm^{-3})^2 = \frac{1}{N}\Sigma_1^N(\text{Model} - \text{Obs})^2 \,, \tag{11}$$

$$\text{RMSE } (\mu gm^{-3}) = \sqrt{\frac{1}{N}\Sigma_1^N(\text{Model} - \text{Obs})^2} \,, \tag{12}$$

245 $$\text{IOA} = 1 - \frac{\Sigma_1^N(\text{Model}-\text{Obs})^2}{\Sigma_1^N(|\text{Model}-\overline{\text{Ob}}|+|\text{Obs}-\overline{\text{Ob}}|)^2} \,, \tag{13}$$

$$R = \frac{\Sigma(\text{Model}-\overline{\text{Mode}})\times(\text{Obs}-\overline{\text{Ob}})}{\sqrt{\Sigma(\text{Model}-\overline{\text{Mode}})^2\times\Sigma(\text{Obs}-\overline{\text{Obs}})^2}} \,, \tag{14}$$

The AQI for PM2.5 was classified into four categories based on the PM2.5-concentration standards used in South Korea. PM2.5 concentrations between 0 $\mu gm^{-3}$ to 15 $\mu gm^{-3}$ were classified as "good"; 16 $\mu gm^{-3}$ to 35 $\mu gm^{-3}$, "moderate"; 36 $\mu gm^{-3}$ to 75 $\mu gm^{-3}$, "bad"; and 76 $\mu gm^{-3}$ or higher, "very bad". The ACC determined the categorical prediction accuracy of the model pertaining to the four AQI 250 categories, and the probability of detection (POD) determined the prediction performance of the model for high PM2.5 concentrations ("bad" and "very bad" AQI categories). The FAR determined the rate of incorrect predictions when the observations tended to be "moderate" or "good" but the predictions pointed to high concentrations ("bad" or "very bad" AQI categories). A low FAR value indicated

better performance. The F1-score indicator, which is the harmonic mean of the POD and FAR, reflected the POD as well as FAR evaluations. Additionally, the recall and precision were evaluated for four categories. The recall is an indicator of how well the model reproduced the categories that appear in observation. The precision is the accuracy that matches the category of observation among the prediction results of the model for each category. Equations (S1) to (S8) were used for calculating the recall and precision. Equations (15) to (18) were used for calculating the AQI prediction-evaluation indicators, and Table 4 lists the intervals corresponding to the four categories for calculating ACC, POD, FAR, recall and precision.

$$\text{ACC (\%)} = \frac{(a1+b2+c3+d4)}{N} \times 100 , \tag{15}$$

$$\text{POD (\%)} = \frac{(c3+c4+d3+d\ )}{(a3+a4+b3+b4+c3+c4+d3+d4)} \times 100 , \tag{16}$$

$$\text{FAR (\%)} = \frac{(c1+c2+d1+d2)}{(c1+c2+c3+c4+d1+d2+d3+d4)} \times 100 , \tag{17}$$

$$\text{F1-score} = 2 \times \frac{\text{POD} \times (100-\text{FAR})}{\text{POD}+(100-\text{FAR})} , \tag{18}$$

Table 4. Intervals corresponding to the four categories for calculating ACC, POD, FAR and recall, precision: "good" ($PM_{2.5} \leq 15$ µgm$^{-3}$), "moderate" (16 µgm$^{-3} \leq PM_{2.5} \leq 35$ µgm$^{-3}$), "bad" (36 µgm$^{-3} \leq PM_{2.5} \leq 75$ µgm$^{-3}$), and "very bad" (76 µgm$^{-3} \leq PM_{2.5}$).

| Level | | Model forecast | | | |
| --- | --- | --- | --- | --- | --- |
| | | Good | Moderate | Bad | Very bad |
| Observation | Good | a1 | b1 | c1 | d1 |
| | Moderate | a2 | b2 | c2 | d2 |
| | Bad | a3 | b3 | c3 | d3 |
| | Very bad | a4 | b4 | c4 | d4 |

The effect of the training data on the prediction performance of the DNN model was quantitatively analyzed using the RMSE indicator. The overall effect of the forecast data on model predictions was calculated based on the RMSE-difference between the DNN-ALL and DNN-OBS models. The effect of the predicted weather data on model predictions was calculated based on the RMSE-difference between the DNN-OPM and DNN-OBS models (Eq. (19)). The effect of the predicted $PM_{2.5}$ data on model predictions was calculated based on the RMSE-difference between the DNN-ALL and DNN-OBS models (Eq. (20)).

$$\text{Contribution of predicted weather (\%)} = \left|\frac{(\text{DNN-OPM}_{D+i})-(\text{DNN-OBS}_{D+i})}{(\text{DNN-ALL}_{D+i})-(\text{DNN-OBS}_{D+i})}\right| \times 100 \ (i = 0 \text{ to } 2) , \tag{19}$$

$$\text{Contribution of predicted PM}_{2.5} \text{ (\%)} = \left|\frac{(\text{DNN-A}_{D+i})-(\text{DNN-OPM}_{D+i})}{(\text{DNN-ALL}_{D+i})-(\text{DNN-O}_{D+i})}\right| \times 100 \ (i = 0 \text{ to } 2) , \tag{20}$$

## 4 Evaluation of prediction performance

The evaluations based on statistics and AQI classifications were conducted for each of the DNN-model experiments (DNN-OBS, DNN-OPM, and DNN-ALL), and the results were compared with those of the CMAQ model currently operational in South Korea. In Sect. 4.1, we examine the daily prediction performance of the three DNN-model experiments and CMAQ model using statistical indicators for the three-day period (D+0 to D+2), and quantitatively analyze the effect of different training data combinations on the prediction performance

of the DNN model. A comparative evaluation with the CMAQ model was conducted to assess whether the DNN-ALL model was more comprehensive for 6-hour average forecasting than the existing daily average forecasting model. In Sect. 4.2, to assess the potential of DNN-ALL as a superior forecasting model, the daily AQI predictions therein for the three-day period (D+0 to D+2) were compared to those of the CMAQ model.

## 4.1 Evaluation of daily prediction performance based on the training data

Table S3 in the Supplement shows the statistical evaluation results of three DNN-model experiments (DNN-OBS, DNN-OPM, and DNN-ALL) and CMAQ model during the training period from 2016 to 2018. In D+0 to D+2, the DNN-ALL model performs the best in terms of all statistical indicators. In addition, the values of all three experiments indicate a decrease in the RMSE compared to the CMAQ model. Table S4 in the Supplement presents the statistical evaluation results of the three experiments and CMAQ models during the validation period in 2019. The DNN-OBS model shows similar performance for D+0 compared to the CMAQ model but decreased performance owing to an increased RMSE of D+1 and D+2 by 2 $\mu gm^{-3}$ and 2.2 $\mu gm^{-3,}$ respectively. The DNN-OPM model shows an increase in performance owing to a decrease in the RMSE of D+0 by 3 $\mu gm^{-3}$ compared to the CMAQ model. Moreover, the RMSE of D+1 and D+2 decrease by 0.4 $\mu gm^{-3}$ and 0.4 $\mu gm^{-3}$ compared to the CMAQ model, respectively, indicating that the performance is similar. For the DNN-ALL model, the RMSE from D+0 to D+2 decreased by 4.6 $\mu gm^{-3}$, 2.7 $\mu gm^{-3}$, and 2.1 $\mu gm^{-3}$, compared to the CMAQ model, which shows an improved performance.

Table 5 summarizes the results of the statistical evaluations of the prediction performances of the three DNN-model experiments and the CMAQ model in the test set (January to March 2021). Figure 6 depicts the corresponding Taylor diagrams, and Figure 7 illustrates the corresponding time series. For D+0, the CMAQ model RMSE was 11.4 $\mu gm^{-3}$ with a 0.90 IOA, and that of the DNN-OBS was 10.8 $\mu gm^{-3}$ with a 0.86 IOA, thereby indicating a lower error and IOA compared to those of the CMAQ model. The RMSEs of the DNN-OPM and DNN-ALL were 8.0 $\mu gm^{-3}$ and 7.3 $\mu gm^{-3}$, respectively, and their IOAs were 0.93 and 0.95, respectively, indicating decreased errors and increased IOAs compared to those of the CMAQ model. Based on the RMSE and IOA values, the DNN-ALL exhibited the best prediction performance. The Taylor diagram (Fig. 6 (a)), which depicts the RMSE, R, and standard deviation indicators simultaneously, confirms that DNN-ALL demonstrated the best prediction performance among the evaluated models. Fig. 7(a1) and 7(a2) reveal that all the three DNN-model experiments exhibited improved overprediction performance compared to the CMAQ model; however, the DNN-OBS exhibited the highest underprediction of PM$_{2.5}$ concentration during the high-concentration period (February 11 to February 14). The domestic and foreign contributions to the high-concentration period were analyzed using the CMAQ with brute-force method (CMAQ-BFM) model (Bartnicki, 1999; Nam et al., 2019). The BFM revealed that the foreign contribution to the PM$_{2.5}$ concentration because of the long-term transport of pollutants to the Seoul area was 68% on February 11, 54% on February 12, 66% on February 13, and 41% on February 14. This aspect of the high PM$_{2.5}$ concentration could not be characterized solely by using observation data (data observed at each point) as the training data. This phenomenon seemed to cause an increase in the concentration on the day subsequent to the day a high concentration occurred. The DNN-OBS RMSE obtained on excluding the high-concentration period was 9.4 $\mu gm^{-3}$, which was lower than that of the CMAQ model (10.9 $\mu gm^{-3}$) and 1.4 $\mu gm^{-3}$ lower than that exhibited by the DNN-OBS model when the high-concentration period was included. In contrast, the RMSEs of the DNN-OPM and DNN-ALL were 7.3 $\mu gm^{-3}$ and 7.0 $\mu gm^{-3}$, respectively, the IOAs were 0.93 and 0.94, respectively, and the R-values were 0.89 for both models, when the high-concentration period was excluded. No significant difference in results was observed even on inclusion of the high-concentration period (February 11 to February 14). These results suggest that when the observation and prediction data are used as the training data, the DNN model reflects the characteristics of the high-

concentration phenomenon caused by long-distance transport. Excluding the high PM$_{2.5}$ concentration caused by long-term transport, the DNN model demonstrated a marginally improved prediction performance compared to the CMAQ model on D+0, even when using only the observation data as the training data. In addition, the use of the prediction data as the training data facilitated an improved prediction performance concerning the long-term-transport-induced phenomenon compared to that of the CMAQ model.

For D+1 and D+2, the CMAQ model RMSEs were 11.2 μgm$^{-3}$ and 13.6 μgm$^{-3}$, respectively, and the IOAs were 0.90 and 0.85, respectively. In contrast, the DNN-OBS RMSEs for D+1 and D+2 were 16.2 μgm$^{-3}$ and 16.9 μgm$^{-3}$, respectively, and the IOAs were 0.44 and 0.27, respectively. Thus, the DNN-OBS model resulted in larger errors and smaller IOAs compared to the CMAQ model. The errors increased and the IOAs decreased for the DNN-OPM, when compared to those of the CMAQ model. However, the DNN-OPM model RMSEs decreased by 4.0 μgm$^{-3}$ and 2.9 μgm$^{-3}$, and the IOAs increased by 0.34 and 0.45 compared to those of the DNN-OBS model, for D+1 and D+2, respectively. The DNN-ALL model performed the best, with RMSEs of 9.0 μgm$^{-3}$ and 10.6 μgm$^{-3}$ and IOAs of 0.90 and 0.86 for D+1 and D+2, respectively, exhibiting smaller errors and larger IOAs compared to those of the CMAQ model. The standard deviations of the DNN-ALL model were 13.5 μgm$^{-3}$ and 12.7 μgm$^{-3}$ for D+1 and D+2, respectively. For D+1 and D+2, DNN-ALL outperformed the remaining DNN-models and the CMAQ model (Fig. 6(b) and 6(c)). This was concluded based on the superior RMSE and R-values exhibited therein. Moreover, as shown in Fig. 7 (b1), (b2), (c1), and (c2), the DNN-ALL model exhibited lower overprediction compared to that by the CMAQ model. However, the DNN-OBS and DNN-OPM models overpredicted low PM$_{2.5}$ concentrations and underpredicted high PM$_{2.5}$ concentrations, when compared to the observation data. The DNN-OBS model did not predict the change in the observed PM$_{2.5}$ concentration after D+0, indicating a decrease in IOA and a limited range of predicted PM$_{2.5}$ concentrations with respect to the observations. Although the DNN-OPM model outperformed DNN-OBS, it was inferior to DNN-ALL because the DNN-OPM training data lacked sufficient features for predicting the change in the observed PM$_{2.5}$ concentration. The DNN-ALL model outperformed the CMAQ model for D+1 and D+2, while all three DNN-based models outperformed the CMAQ model for D+0. For D+1 and D+2, the RMSE of the DNN-ALL model using the prediction data from numerical models decreased by 7.2 μgm$^{-3}$ and 6.3 μgm$^{-3}$, respectively, compared to DNN-OBS. The effects of weather forecast data were 56% (4 μgm$^{-3}$) and 46% (2.9 μgm$^{-3}$), respectively, and those of predicted PM$_{2.5}$ concentration were 44% (3.2 μgm$^{-3}$) and 54% (3.4 μgm$^{-3}$), respectively, when used as training data. These results suggest that as the prediction period lengthens, the weather forecast and PM$_{2.5}$ concentration prediction data are more important than current observation data for improving the model prediction performance.

Also, the performance of the Random Forest (RF) model, one of the statistical models, was evaluated and compared with DNN-ALL. Table S5 in the Supplement shows of the statistical evaluation of the Random Forest (RF) and DNN-ALL model shown the best results in the statistical evaluation at the three experiments and CMAQ model. Compared to the RF model, the RMSE value of the DNN-ALL model decreased by 0.6 to 1.9 μgm$^{-3}$, and the R and IOA values increased slightly. Although the volume of training data in this paper was not sufficiently huge to be applied to DNN model, the DNN model outperformed the RF model. In the future, DNN model can also reflect the expansion of training data and consider the scalability of the model that can predict future data growth over time and segmentation with a 1-h interval. Therefore, the performance of the DNN model is expected to improve as the training data increases.

In modern times, people demand the availability of more detailed forecasts, well in advance of the average daily forecast, to enable better planning of daily lives and the mitigation of air-polluting emissions. Therefore, the applicability of the DNN-ALL model as a 6-hour forecast model was evaluated. Furthermore, the 6-hour mean prediction performance of DNN-ALL was evaluated against that of the CMAQ model. Table 6 presents the RMSE and IOA for each T-step of the DNN-ALL and CMAQ. The RMSEs of the DNN-ALL ranged between 7.3 μgm$^{-3}$ to 16.0 μgm$^{-3}$, a decrease of 2.7 μgm$^{-3}$ to 8.8 μgm$^{-3}$ compared to the CMAQ model. The DNN-ALL IOAs ranged between 0.74 and 0.97, indicating higher or similar IOAs than those of the CMAQ-model. However, the RMSE and IOA of DNN-ALL not

decrease monotonically. This is because the model performance may differ according to the conditions of target time such as daytime, nighttime, high concentration, and low concentration. As shown in the CMAQ results, the prediction performance of the DNN-ALL model degrades or improves monotonically over time.


Table 5. Statistical summary of daily PM$_{2.5}$ concentration prediction performance of the CMAQ, DNN-OBS, DNN-OPM, and DNN-ALL models.

| Model | Day | MSE (($\mu gm^{-3}$)$^2$) | RMSE ($\mu gm^{-3}$) | R | IOA |
|---|---|---|---|---|---|
| CMAQ | D+0 | 130.4 | 11.4 | 0.83 | 0.90 |
| | D+1 | 125.4 | 11.2 | 0.82 | 0.90 |
| | D+2 | 185.0 | 13.6 | 0.74 | 0.85 |
| DNN-OBS | D+0 | 116.6 | 10.8 | 0.79 | 0.86 |
| | D+1 | 262.4 | 16.2 | 0.31 | 0.44 |
| | D+2 | 285.6 | 16.9 | 0.17 | 0.27 |
| DNN-OPM | D+0 | 64.0 | 8.0 | 0.89 | 0.93 |
| | D+1 | 148.8 | 12.2 | 0.70 | 0.78 |
| | D+2 | 196.0 | 14.0 | 0.59 | 0.72 |
| DNN-ALL | D+0 | 53.3 | 7.3 | 0.91 | 0.95 |
| | D+1 | 81.0 | 9.0 | 0.85 | 0.90 |
| | D+2 | 112.4 | 10.6 | 0.79 | 0.86 |

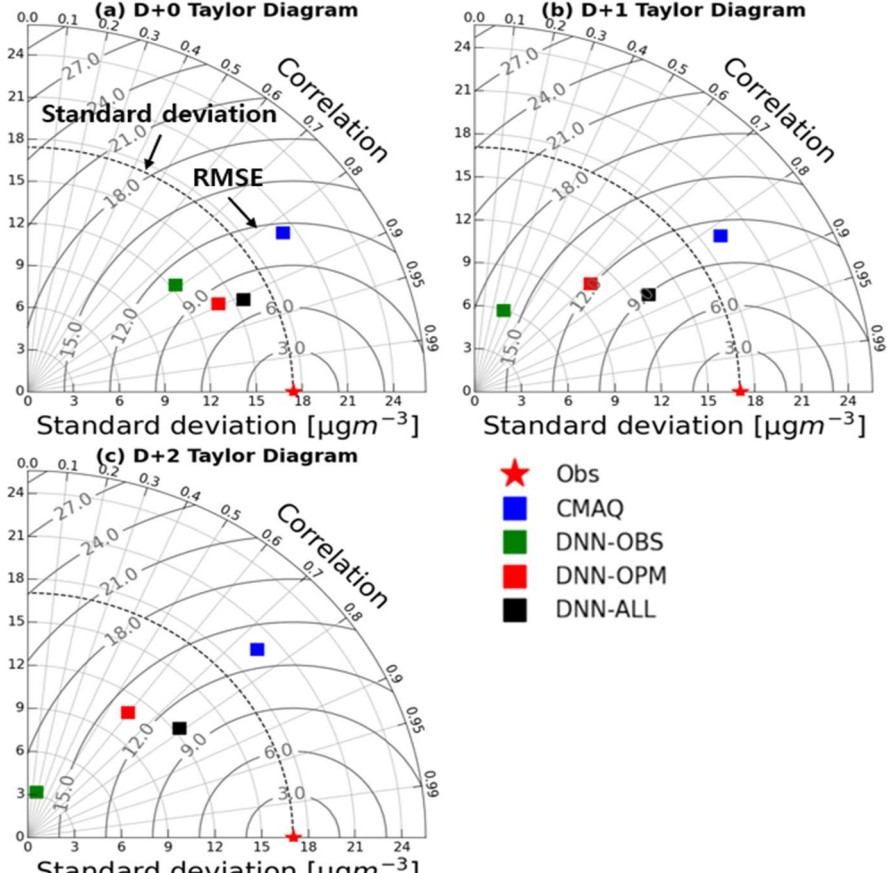

**Figure 6. Taylor diagrams for D+0 to D+2 ((a) to (c)) of the CMAQ, DNN-OBS, DNN-OPM, and DNN-ALL models. In each diagram, the contour line connecting the x- and y-axes represents the standard deviation, and the dark gray contour line represents the RMSE. The smaller the RMSE, the higher the R value; the closer the standard deviation is to the standard deviation of the observation, the closer it is to the Obs (★).**


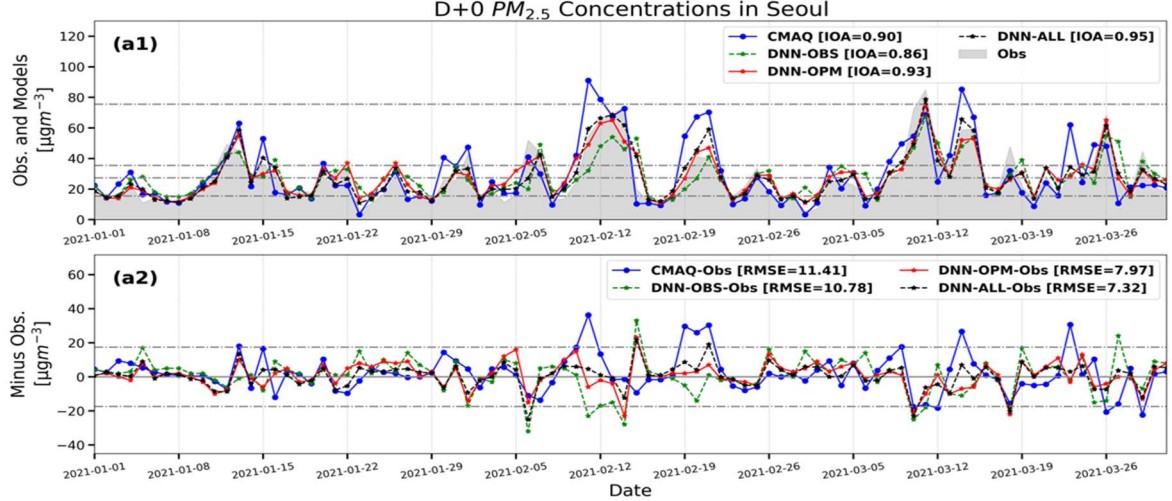

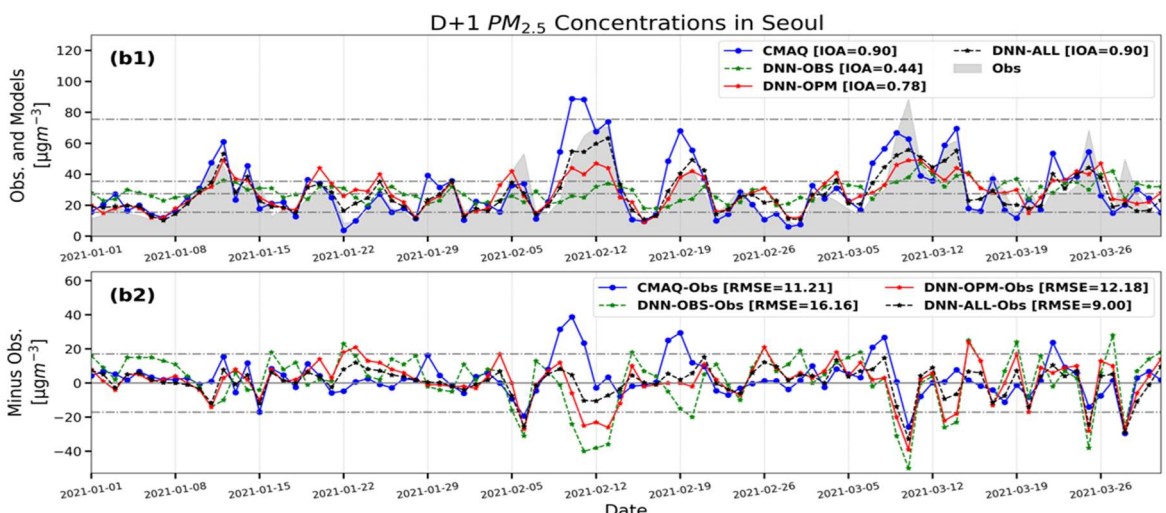

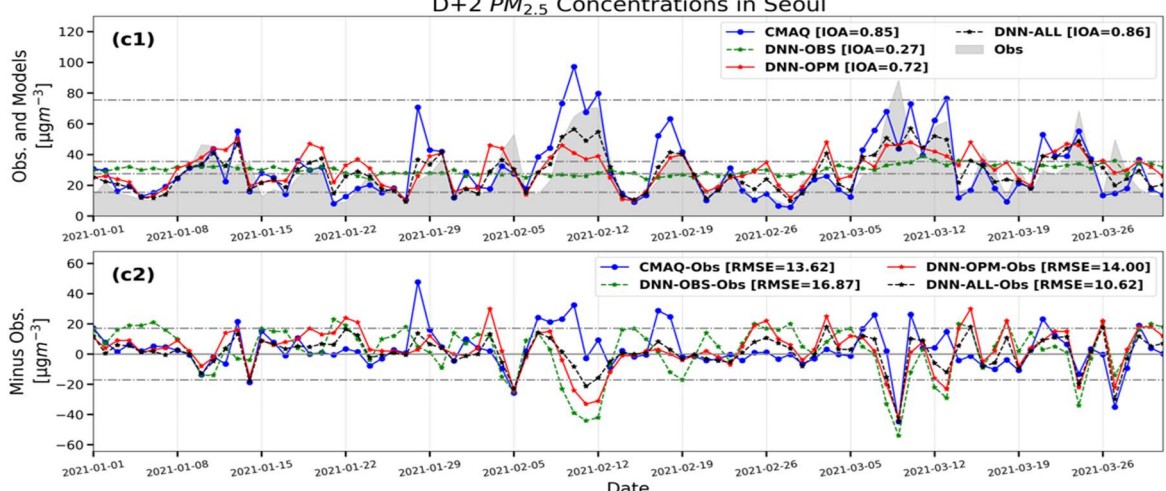

**Figure 7. Time series of PM<sub>2.5</sub> concentrations from observations and predictions using the CMAQ, DNN-OBS, DNN-OPM and DNN-ALL. 7 (a1) to (c1) depict the time series of predictions and observations and (a2) to (c2) depict the differences between the predictions and observations (predictions minus observations). In 7 (a1) to (c1), each of the dashed lines represents values of 15.5 μgm⁻³, 35.5 μgm⁻³, 75.5 μgm⁻³, and average value of observation (27 μgm⁻³). In 7 (a2) to (c2), the dashed lines represent the standard deviation of observation PM₂.₅ as negative and positive.**

Table 6. Statistical summary of the performances of the CMAQ and DNN-ALL models in case of 6-hour average PM₂.₅ forecasts.

| Model | Indicator | T-step | | | | | | | | | | |
|---|---|---|---|---|---|---|---|---|---|---|---|---|
| | | $T_{12\_D0}$ | $T_{18\_D0}$ | $T_{24\_D0}$ | $T_{06\_D1}$ | $T_{12\_D1}$ | $T_{18\_D1}$ | $T_{24\_D1}$ | $T_{06\_D2}$ | $T_{12\_D2}$ | $T_{18\_D2}$ | $T_{24\_D2}$ |
| CMAQ | RMSE (μgm⁻³) | 16.1 | 14.2 | 16.5 | 18.1 | 16.9 | 12.9 | 15.3 | 19.0 | 16.6 | 18.5 | 16.3 |
| | IOA | 0.85 | 0.85 | 0.82 | 0.80 | 0.84 | 0.88 | 0.84 | 0.78 | 0.84 | 0.75 | 0.82 |
| DNN-ALL | RMSE (μgm⁻³) | 7.3 | 9.0 | 12.4 | 14.5 | 13.4 | 10.2 | 12.3 | 16.0 | 13.5 | 13.9 | 13.6 |
| | IOA | 0.97 | 0.92 | 0.86 | 0.83 | 0.86 | 0.87 | 0.86 | 0.77 | 0.85 | 0.74 | 0.80 |

## 4.2 AQI-prediction performance

Among the three experiments described in Sect. 4.1, the DNN-ALL model demonstrated the best results in the statistical evaluation. The AQI-prediction performance of the DNN-ALL model was compared with that of the CMAQ and RF model.

Table 7 and Fig. 8 present the AQI evaluation results of the DNN-ALL and CMAQ models. The overall ACC of the DNN-ALL model for D+0 was 77.8%, approximately 12% higher than that of the CMAQ model. The categorical-prediction ACC of the DNN-ALL was greater than that of the CMAQ model by approximately 7% for "good", 17% for "moderate", 4% for "bad", and 100% for "very bad". During the target period of this study, "very bad" occurred once. Although DNN-ALL predicted this occurrence accurately, the CMAQ predicted "bad", indicating a 100% difference in accuracy between the two models (Fig. 8 (a1), (b1)). The F1-score was 80%, 3% higher than that of the CMAQ model. The FAR of the DNN-ALL model improved by approximately 17%, although the POD decreased by approximately 9%.

These results suggest that the DNN-ALL model overpredicted less than the CMAQ model, whose predicted PM$_{2.5}$ concentrations were

generally higher than the observed values.

For D+1 and D+2, the overall ACC was 64.6% and 61.1%, respectively, an approximate decrease of 2% and 1%, respectively, compared to the CMAQ model. The AQI-prediction ACC of the DNN-ALL model decreased by approximately 27% on both days in "good", and increased by approximately 12% for D+1 and 5% for D+2 in "moderate". The "good" ACC was low because the CMAQ model underpredicted, and the DNN-ALL overpredicted, with respect to the observed values. An equal "bad" ACC of 70% was obtained via DNN-ALL and CMAQ for D+1,

which increased by 20% for the DNN-ALL model on D+2 (Fig. 8(a2), 8(a3), 8(b2), and 8(b3)). The F1-scores of DNN-ALL and CMAQ for D+0 were 70% and 67%, respectively; however, the F1-score increased for DNN-ALL by 1% for D+1 and 7% for D+2. For the DNN-ALL model, in case of D+1, the POD decreased by 10% and FAR improved by 8%p, whereas, in case of D+2, the POD increased by 5% and FAR improved by 8%.

Table S6 in the Supplement shows the precision and recall of all categories for the DNN-ALL and CMAQ models. The precision and recall of

the DNN-ALL model in the bad category are presented to be higher than those of the CMAQ model. In the bad category of D+0, the precision and recall of DNN-ALL are greater than those of the CMAQ model by 0.24 and 0.04, respectively. In addition, in the very bad category, the precision and recall of DNN-ALL are to be 1.0 equally higher than those of the CMAQ model. In D+1, the precision of DNN-ALL in the bad category is greater than that of the CMAQ model by 0.1, but the recall is similar to the CMAQ model. In D+2, the precision and recall for the bad category of DNN-ALL increased by 0.14 and 0.20 compared to the CMAQ model, respectively. These

results show that the performance of the DNN-ALL model is superior to that of the CMAQ model for predicting high concentrations that affect the health of the people.

Table S7 in the Supplement shows the AQI evaluation results of the DNN- ALL and RF models. The ACC of the DNN-ALL model increased by approximately 2 to 13 %p compared to the RF model, and the F1-score decreased by 1 %p at D+1 1 but increased by 1 %p and 9 %p at D+0 and D+2, respectively.


Table 7. Categorical forecast scores of the performance of the CMAQ and DNN-ALL models.

| Model | Day | ACC (%) | | POD (%) | | FAR (%) | | F1-score (%) |
|---|---|---|---|---|---|---|---|---|
| CMAQ | D+0 | 65.6 | 59/90 | 81.8 | 18/22 | 28.0 | 7/25 | 77 |
| | D+1 | 66.7 | 60/90 | 81.0 | 17/21 | 39.3 | 11/28 | 69 |
| | D+2 | 62.2 | 56/90 | 71.4 | 15/21 | 48.3 | 14/29 | 60 |
| DNN-ALL | D+0 | 77.8 | 70/90 | 72.7 | 16/22 | 11.1 | 2/18 | 80 |
| | D+1 | 64.4 | 58/90 | 71.4 | 15/21 | 31.8 | 7/22 | 70 |
| | D+2 | 61.1 | 55/90 | 76.2 | 16/21 | 40.7 | 11/27 | 67 |

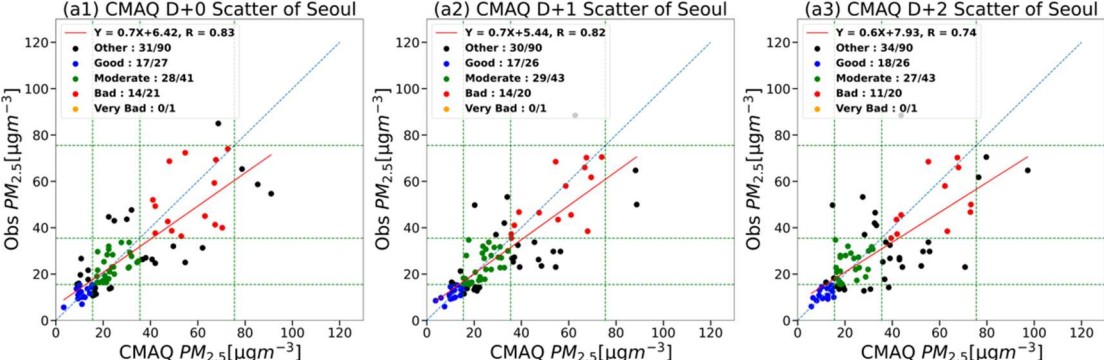

**(a) Scatter plot of the CMAQ model for D+0 to D+2**

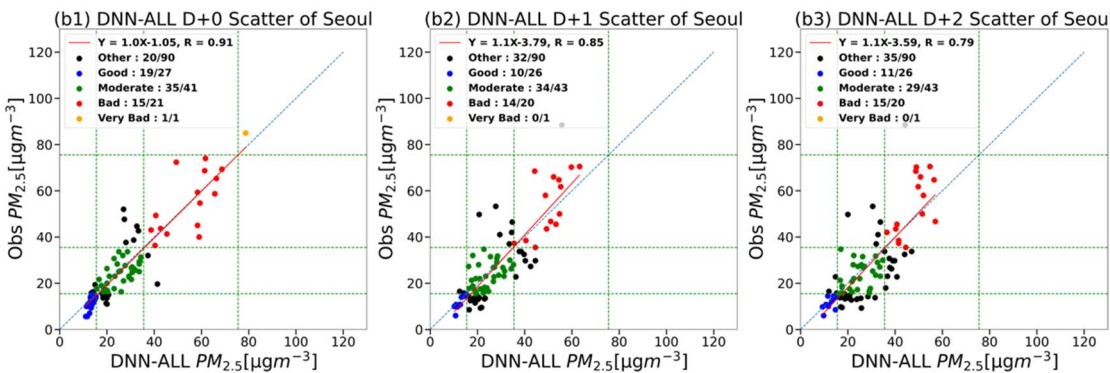

**(b) Scatter plot of the DNN-ALL model for D+0 to D+2**

**Figure 8. Observations from D+0 to D+2 and corresponding scatter plots of the DNN-ALL and CMAQ models. The blue dots indicate the observation and prediction values in the AQI category "good"; the green dots, "moderate"; the red dots, "bad"; the orange dots, "very bad."**


# 5 Conclusion

This study aimed to develop a deep neural network (DNN) model for predicting the 6-hour average $PM_{2.5}$ concentration for three days (D+0 to D+2) using the DNN algorithm based on observation, weather forecast, and $PM_{2.5}$ concentration forecast data. The prediction performance of the DNN model was comparatively evaluated against that of the CMAQ model currently used to forecast air quality in South Korea. The effects of different training data on the prediction performance of the DNN model were also analyzed.

For D+0, the DNN-ALL, DNN-OPM and DNN-OBS models exhibited RMSE decreases of 4.1 $\mu gm^{-3}$, 2.8 $\mu gm^{-3}$, and 0.6 $\mu gm^{-3}$, respectively, and similar IOA values, compared to the CMAQ model, thereby indicating improved performance. For D+1 and D+2, the prediction performance of the DNN-ALL model was the best, with RMSE decreases (owing to lower overprediction) compared to those in the CMAQ model of 2.2 $\mu gm^{-3}$ and 3.0 $\mu gm^{-3}$, for D+1 and D+2, respectively. In contrast, the DNN-OBS performed poorly compared to the CMAQ model, with RMSE-increases of 2.8 $\mu gm^{-3}$ and 3.3 $\mu gm^{-3}$ and sharp IOA-decreases of 0.46 and 0.58, for D+1 and D+2, respectively. The DNN-OPM prediction performance was marginally inferior to that of the CMAQ, with RMSE increases of 1.0 $\mu gm^{-3}$ for D+1 and 0.4 $\mu gm^{-3}$ for D+2. The RMSE decrease in case of DNN-ALL was 7.2 $\mu gm^{-3}$ for D+1 and 6.3 $\mu gm^{-3}$ for D+2, compared to DNN-OBS, indicating that the use of forecasting data as the training data greatly affected the performance of the DNN model pertaining to longer forecasting periods. The RMSE of the DNN-ALL decreased within a range of 2.7 $\mu gm^{-3}$ to 8.8 $\mu gm^{-3}$ compared to the CMAQ model in case of the 6-hour average prediction, implying that the DNN model could perform better than the CMAQ in both, 6-hour average and daily forecasting. The F1-score of the DNN-ALL improved by 3%, 1% and 7%, and false alarms decreased by 17%, 8% and 8% compared to the CMAQ model for each day. These results demonstrate the better prediction ability of the DNN model in case of high $PM_{2.5}$ concentrations, as it rendered fewer false alarms by decreasing overpredictions, unlike the CMAQ model. Thus, the evaluation results reveal that the DNN model could be useful 6-hour average and daily forecasts.

For further performance-improvement of the DNN model, spatial training data should be expanded to reflect the changes in $PM_{2.5}$ concentration induced by the surrounding areas, and the training duration should be increased to allow learning pertaining to the varying concentrations. In addition, the improvement of the numerical models used for generating weather and air-quality prediction data is necessary.

When high $PM_{2.5}$ concentrations are predicted, mitigation policies are implemented for the protection of public health in South Korea. These policies aim to reduce air-polluting emissions by limiting the power-generation capacity of thermal power plants and operation of vehicles, which are processes that involve socio-economic costs. Consequently, inaccurate forecasts of high $PM_{2.5}$ concentrations can result in socio-economic losses. Therefore, the use of the DNN model for forecasting is expected to reduce economic losses and protect public health.

## Code availability

The code and data used in this study can be found at http://doi.org/10.5281/zenodo.5652289 or https://github.com/GercLJB/GMD.

## Supplement

The supplement related to this article is available online at : .

## Author contributions

LJB (Lee Jeong-Beom) wrote the manuscript and contributed to the DNN model development and optimization. LJB (Lee Jae-Bum) supervised this study, contributed to the study design and drafting, and served as the corresponding author. KYS and KHY contributed to the generation of the training data for the DNN model. CMH, PHJ, and LDG contributed to the real-time operation of the CMAQ model.

## Competing interests

The authors declare no conflicts of interest.

## Disclaimer

Publisher's note: Copernicus Publications remains neutral with regard to jurisdictional claims in published maps and institutional affiliations.

## Acknowledgements

This study was conducted with the support of the Air Quality Forecasting Center at the National Institute of Environmental Research under the Ministry of Environment (NIER-2021-01-01-086).

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
