# Peer review of "Development of a deep neural network for predicting 6-hour average $PM_{2.5}$ concentrations up to two subsequent days using various training data"

_Geoscientific Model Development, 2021_

## Referee Comment (RC1)

This paper addresses a relevant and important topic related to improved air quality forecasting. It uses deep neural networks for that and applies to data from a single air quality station in Seoul. Three different experimental configurations are included, namely forcing with observed air quality measurements only, forecasting with observed air quality and forecasted weather data, and forecasting with observed air quality, weather, and predicted air quality from a physics model.

There are a couple of issues with the paper that hinders understanding:

1) It's not clear what the contributions of the paper are versus what already exists. Did the authors run the WRF and CMAQ models to generate the training data or were these data obtained from some other source?

2) Similarly there is no justification for the choice of model. I would like to see the approach benchmarked against simpler models (ARIMA, Random Forest, … basically anything from the statistical or machine learning family to compare against the deep learning approach). The volume of data that the model is trained on are not huge so it is not apparent that a DNN is the best choice of algorithm

3) Many details on the DNN model setup are presented with no real justification for their choice e.g. using the membership function for temporal features, the choice of DNN architecture such as number of layers is not explained. On line 50 – 55 the authors 1) note the advantages of RNN for time series forecasting and 2) that Kim et al. (2019) developed an RNN model to predict PM2.5 concentrations at two locations in Seoul. Why was RNN not considered for this study rather than DNN and how does the performance of this model compare to that reported by Kim et. al. Similarly it is not clear if the autoregressive features of the data were expressed in any form? Of course RNN expresses these implicitly but in other models it can be advantageous to feature engineer the autoregressive dependencies. Were any feature combinations other than those reported explored in the paper. The authors need to justify the choice of algorithm and how the DNN was designed detailing such information as feature selection, number of layers/nodes.

4) The manuscript could be improved to enhance readability and replicability of the study. I appreciate the authors making code and data available on Zenodo. I would however encourage them to create a GitHub repository with some documentation to allow people easily replicate the results. As mentioned in 1) authors could be more descriptive when detailing data sources. Some parts of the paper could be explained better, e.g. line 161 "average weather and air quality prediction data". What is meant by average here? Spatial or temporal. Are WRF and CMAQ data extracted from the entire Seoul area domain or subset corresponding to location of the observation point. Line 166 – 167 "ensure that the training data were not biased" – feature scaling does not ensure unbiased datasets, it simply helps the model learn better. The data could still be biased. Line 173: "undergoes feature scaling through the backpropagation algorithm" – not clear what is meant by feature scaling in this context. Line 128 "16 meteorological forecast variables were created by the WRF model" – I believe what is meant here is that 16 variables were extracted as features but many more variables were generated by the WRF model.

5) I really don't see the relevance of section 4.2. The models have already been compared and evaluated in terms of predictive skill in regression. Then you take the same models and evaluate in terms of a classification model but only whether they

predicted within those bounds (i.e. the model and results are the same the only thing that changes are the interpretation)

Other more minor comments:
1) What is the membership function defined in line 144? Is this the generation of temporal features described in subsequent lines? Why was time data encoded in this manner? It seems more standard to represent as integer values or to convert those integer values to cyclic features (i.e. so that month 12 and month 1 are close to each other rather than far away). I haven't seen this approach used previously and would like to understand the motivation and/or justification.
2) The test period is quite short – 3 months out of 51 months. Was there a reason for this?
3) Line 159 – 163: This is a quite confusing way to present forecast horizons. I'd suggest to just use hours and present forecast horizons as T06, T12, T18, T24, ... Mixing days + hours and having different chunks within each day is confusing to the reader.
4) Line 167 – 168: I don't quite understand why data was both standardised and normalised? Did this improve performance versus just using normalisation (if you wished to have bounded between 0 and 1) or indeed versus the unscaled data? Generally people chose either standardisation or normalisation so I'm curious why you did both
5) Figure 4: What does Epoch_n = Epoch_n-1 + 1 mean? What does Epoch_n-1 of validation cost > Epoch_n of validation cost mean? Should it be Validation cost of Epoch_n?
6) Equation 11 and 12, I'm not sure the use of both MSE and RMSE is necessary and could probably drop one.
7) Line 225 – 230: In classification problems, accuracy, precision and recall are the standard metrics presented. I would suggest in this paper you also include (you already do accuracy so I suggest adding precision and recall)
8) Line 280 – 281: This is a difficult narrative to support. You are using the CMAQ model output as the training data and then saying the usage of that training data allows the model to better represent the long-term-transport-induced phenomenon.
9) What do the dashed lines in the residual figures represent in Figure 7.

---

## Author Comment (AC1)

We thank the editor and the reviewers for the time and effort put in towards the review of this manuscript. The insightful comments and suggestions have helped improve the manuscript significantly. We have incorporated several changes based on the suggestions of the reviewers. The detailed responses to the reviewers' comments are given below.

**Section 1. Major comments.**

**Question 1.** It's not clear what the contributions of the paper are versus what already exists. Did the authors run the WRF and CMAQ models to generate the training data or were these data obtained from some other source?

**Answer 1.** We understand the concern raised by the reviewer. We directly generated the training data using the WRF and CMAQ. We will make revisions to the manuscript to clarify this point.

**Question 2.** Similarly there is no justification for the choice of model. I would like to see the approach benchmarked against simpler models (ARIMA, Random Forest, … basically anything from the statistical or machine learning family to compare against the deep learning approach). The volume of data that the model is trained on are not huge so it is not apparent that a DNN is the best choice of algorithm.

**Question 2-1**. Similarly, there is no justification for the choice of model. I would like to see the approach benchmarked against simpler models (ARIMA, Random Forest, … basically anything from the statistical or machine learning family to compare against the deep learning approach).

**Answer 2-1.** Among the statistical models mentioned by the reviewer, the prediction performances of the Random Forest (RF) have been evaluated and compared with that of the DNN-ALL. Table 1 shows the results of the statistical evaluations, and Table 2 lists the results of the Air Quality Index (AQI) evaluation. Compared to the results of the RF model, the Root Mean Square Error (RMSE) value of the DNN-ALL model decreased by 0.6 to 1.9 $\mu gm^{-3}$, and the Correlation Coefficient (R) and Index of Agreement (IOA) values increased slightly. The Accuracy (ACC) of the DNN-ALL model increased by approximately 2–13 %p compared to the RF model, and the F1-score decreased by 1 %p at D+1 but increased by 1 %p and 9 %p at D+0 and D+2, respectively. A comparison of the performance results showed that the DNN-ALL model outperformed the RF model. We will this information in the revised manuscript to include these results.

Table 1. Statistical performance of the DNN-ALL and Random Forest models.

| Model | Day | MSE $((\mu gm^{-3})^2)$ | RMSE $(\mu gm^{-3})$ | R | IOA |
|---|---|---|---|---|---|
| DNN-ALL | D+0 | 53.3 | 7.3 | 0.91 | 0.95 |
| | D+1 | 81.0 | 9.0 | 0.85 | 0.90 |
| | D+2 | 112.4 | 10.6 | 0.79 | 0.86 |
| Random Forest | D+0 | 62.4 | 7.9 | 0.90 | 0.93 |
| | D+1 | 106.1 | 10.3 | 0.83 | 0.85 |
| | D+2 | 156.3 | 12.5 | 0.73 | 0.76 |

Table 2. Categorical performance of the DNN-ALL and Random Forest model.

| Model | Day | ACC (%) | | POD (%) | | FAR (%) | | F1-score (%) |
|---|---|---|---|---|---|---|---|---|
| DNN-ALL | D+0 | 77.8 | 70/90 | 72.7 | 16/22 | 11.1 | 2/18 | 80 |
| | D+1 | 64.4 | 58/90 | 71.4 | 15/21 | 31.8 | 7/22 | 70 |
| | D+2 | 61.1 | 55/90 | 76.2 | 16/21 | 40.7 | 11/27 | 67 |
| Random Forest | D+0 | 75.6 | 68/90 | 77.3 | 17/22 | 19.0 | 4/21 | 79 |
| | D+1 | 61.1 | 55/90 | 76.2 | 16/21 | 33.3 | 8/24 | 71 |
| | D+2 | 48.9 | 44/90 | 71.4 | 15/21 | 50.0 | 15/30 | 58 |

**Question 2-2**. The volume of data that the model is trained on are not huge so it is not apparent that a DNN is the best choice of algorithm.

**Answer 2-2.** We agree that the volume of data in this paper is not sufficiently huge to be applied to artificial intelligence (AI). Nevertheless, the reason for choosing DNN algorithm is to take into account the scalability of the model, which can reflect training data expansion to forecast the segmentation with a 1-h interval and the future data growth over time. Therefore, the performance of the AI is expected to improve as the training data increases.

**Question3.** Many details on the DNN model setup are presented with no real justification for their choice e.g. using the membership function for temporal features, the choice of DNN architecture such as number of layers is not explained. On line 50 – 55 the authors 1) note the advantages of RNN for time series forecasting and 2) that Kim et al. (2019) developed an RNN model to predict PM2.5 concentrations at two locations in Seoul. Why was RNN not considered for this study rather than DNN and how does the performance of this model compare to that reported by Kim et. al. Similarly it is not clear if the autoregressive features of the data were expressed in any form? Of course RNN expresses these implicitly but in other models it can be advantageous to feature engineer the autoregressive dependencies. Were any feature combinations other than those reported explored in the paper. The authors need to justify the choice of algorithm and how the DNN was designed detailing such information as feature selection, number of layers/nodes.

**Question 3-1.** Many details on the DNN model setup are presented with no real justification for their choice e.g. using the membership function for temporal features.

**Answer 3-1.** . In this paper, the membership function was used to reflect these monthly change characteristics. As shown in Figure 1 (Figure 5 in the paper), $PM_{2.5}$ concentration in Seoul is high in January, February, March, and December, and low from August to October. $PM_{2.5}$ concentration has a characteristic that changes gradually from month to month. The examples of how membership function is applied are described in lines **151–153** of the paper. The membership function was applied based on the results presented by Yu et al. (2019). Yu et al. (2019) performed training that reflected monthly change characteristics to improve the high-concentration $PM_{10}$ forecast performance. As indicated by the experiment results presented in Table 3, the POD performance of the training model reflecting the characteristics of the monthly change was improved by 25 %p. The information related to this will be added in the paper.

Table 3. Results of artificial intelligence model performance evaluation when using and without the membership function presented in Yu et al. (2019).

| Model | Day | ACC (%) | POD (%) | FAR (%) |
|---|---|---|---|---|
| Using Membership function | D+1 | 70 | 75 | 48 |
| Without Membership function | D+1 | 76 | 50 | 33 |

[Figure]

Figure 1. Time series of the average monthly PM$_{2.5}$ concentrations from 2016 to 2019.

**Question 3-2.** The choice of DNN architecture such as number of layers is not explained. The authors need to justify the choice of algorithm and how the DNN was designed detailing such information as feature selection, number of layers/nodes. Were any feature combinations other than those reported explored in the paper.

**Answer 3-2.** In order to provide the justification for the layer selection mentioned by the reviewer, we presented the evaluation results according to the number of layers. The statistical and AQI evaluation results of the DNN-ALL model based on the layer are presented in Tables 4 and 5, respectively. The results of the 4-layer and 5-layer models show that the performance is similar. However, compared with the 4-layer model, the RMSE of the 5-layer decreases by approximately 0.1 $\mu gm^{-3}$ to 1 $\mu gm^{-3}$ at D+0 to D+2, and the ACC of the 5-layer model increases by approximately 1 %p to 6 %p at D+0 to D+2. Therefore, the 5-layer model shows the best performance. The 6-layer and 8-layer models contain errors that converge without decreasing during the training process of the model (vanishing gradient problem). The authors believe that the cause of this problem is the activate function. Therefore, as the layer becomes deeper, the value of the last output cannot be significantly changed due to the sigmoid function. We will include this information in the revised manuscript.

Table 4. Statistical evaluation results according to the number of layers.

| Model | Day | MSE (($\mu gm^{-3})^2$) | RMSE ($\mu gm^{-3}$) | R | IOA |
|---|---|---|---|---|---|
| 2-layer | D+0 | 59.3 | 7.7 | 0.91 | 0.94 |
| | D+1 | 92.1 | 9.6 | 0.86 | 0.89 |
| | D+2 | 156.3 | 12.5 | 0.75 | 0.80 |

| | D+0 | 54.7 | 7.4 | 0.91 | 0.95 |
|---|---|---|---|---|---|
| 4-layer | D+1 | 88.3 | 9.4 | 0.86 | 0.90 |
| | D+2 | 134.5 | 11.6 | 0.77 | 0.84 |
| 5-layer (DNN-ALL) | D+0 | 53.3 | 7.3 | 0.91 | 0.95 |
| | D+1 | 81.0 | 9.0 | 0.85 | 0.90 |
| | D+2 | 112.4 | 10.6 | 0.79 | 0.86 |
| 6-layer | D+0 | 174.2 | 13.2 | 0.81 | 0.66 |
| | D+1 | 292.4 | 17.1 | 0 | 0.17 |
| | D+2 | 292.4 | 17.1 | 0 | 0.17 |
| 8-layer | D+0 | 302.7 | 17.4 | 0 | 0.15 |
| | D+1 | 292.4 | 17.1 | 0 | 0.17 |
| | D+2 | 292.4 | 17.1 | 0 | 0.17 |

Table 5. AQI evaluation results according to the number of layers.

| Model | Day | ACC (%) | | POD (%) | | FAR (%) | | F1-score (%) |
|---|---|---|---|---|---|---|---|---|
| 2-layer | D+0 | 70.0 | 63/90 | 81.8 | 18/22 | 28.0 | 7/25 | 77 |
| | D+1 | 55.6 | 50/90 | 81.0 | 17/21 | 39.3 | 11/28 | 69 |
| | D+2 | 51.1 | 46/90 | 81.0 | 17/21 | 50.0 | 17/34 | 61 |
| 4-layer | D+0 | 71.1 | 64/90 | 81.8 | 18/22 | 28.0 | 7/25 | 76 |
| | D+1 | 60.0 | 54/90 | 85.7 | 18/21 | 35.7 | 10/28 | 73 |
| | D+2 | 60.0 | 54/90 | 81.0 | 17/21 | 45.2 | 14/31 | 65 |
| 5-layer (DNN-ALL) | D+0 | 77.8 | 70/90 | 72.7 | 16/22 | 11.1 | 2/18 | 80 |
| | D+1 | 64.4 | 58/90 | 71.4 | 15/21 | 31.8 | 7/22 | 70 |
| | D+2 | 61.1 | 55/90 | 76.2 | 16/21 | 40.7 | 11/27 | 67 |
| 6-layer | D+0 | 55.6 | 50/90 | 50 | 11/22 | 8.3 | 1/12 | 64 |
| | D+1 | 47.8 | 43/90 | 0 | 0/21 | 0 | 0/0 | 0 |
| | D+2 | 47.8 | 43/90 | 0 | 0/21 | 0 | 0/0 | 0 |
| 8-layer | D+0 | 45.6 | 41/90 | 0 | 0/22 | 0 | 0/0 | 0 |
| | D+1 | 47.8 | 43/90 | 0 | 0/21 | 0 | 0/0 | 0 |
| | D+2 | 47.8 | 43/90 | 0 | 0/21 | 0 | 0/0 | 0 |

**Question 3-3.** On line 50 – 55 the authors 1) note the advantages of RNN for time series forecasting. and 2) that Kim et al. (2019) developed an RNN model to predict PM2.5 concentrations at two locations in Seoul. Why was RNN not considered for this study rather than DNN and how does the performance of this model compare to that reported by Kim et. al.

**Answer 3-3-1. (Reason why RNN was not considered)** There are very few studies and relatively less research to predict air quality using AI such as DNN, RNN and CNN, although it has increased recently. Therefore, the purpose of this study is to evaluate the performance of fine dust prediction when using the DNN among various AI algorithms. The RNN is known to have the advantage of time series prediction, and the DNN is known to have the advantage of extracting characteristics of training data well. There is no convergent result confirming which of the two algorithms is better when applied to fine dust

prediction. Therefore, we first performed the simulation using the DNN rather than the RNN in order to maximize the advantages of the DNN for predicting fine dust. In the future, we plan to perform comparative evaluation with the DNN results presented in this paper through the development of RNN models.

95

**Answer 3-3-2. (Comparison with Kim et al.)** We compared the results obtained by Kim et al. (2019) with those obtained in our study. Kim et al. (2019) performed a $PM_{2.5}$ concentration prediction for two out of 41 measuring stations that are located in the Seoul area. However, in this paper, the average $PM_{2.5}$ concentration prediction for 41 measuring stations in Seoul was performed. In other words, there is a spatial difference for the area to be predicted. In addition, the periods of prediction for

100   the two papers are different. The forecast period considered by Kim et al. (2019) was four months, from January 2016 to April 2016, and the forecast period in this study was three months, from January 2021 to March 2021. Although it is difficult to directly compare the two studies because of the existence of temporal and spatial differences, the results of the prediction performance are presented in Table 6. Because Kim et al. (2019) performed only the D+1 prediction, the comparison of the prediction performance with this paper was conducted for D+1. The values indicate that the RMSE is decreased and the IOA

105   is increased compared to other models.

Table 6. Statistical performance of the DNN-ALL and Random Forest models.

| Model | Day | RMSE ($\mu gm^{-3}$) | IOA |
|---|---|---|---|
| DNN-ALL | D+1 | 9.0 | 0.90 |
| Seoul-1 (Kim et. al (2019)) | D+1 | 12.5 | 0.71 |
| Seoul-2 (Kim et. al (2019)) | D+1 | 15.1 | 0.77 |

**Question 3-4.** Similarly, it is not clear if the autoregressive features of the data were expressed in any form? Of course, RNN

110   expresses these implicitly but in other models it can be advantageous to feature engineer the autoregressive dependencies.
**Answer 3-4.** The RNN algorithm implicitly reflects autoregressive features, but the DNN algorithm does not reflect autoregressive features. This study did not consider any autoregressive features.

115   **Question4.** The manuscript could be improved to enhance readability and replicability of the study. I appreciate the authors making code and data available on Zenodo. I would however encourage them to create a GitHub repository with some documentation to allow people easily replicate the results. As mentioned in 1) authors could be more descriptive when detailing data sources. Some parts of the paper could be explained better, e.g. line 161 "average weather and air quality prediction data". What is meant by average here? Spatial or temporal. Are WRF and CMAQ data extracted from the entire Seoul area domain

120   or subset corresponding to location of the observation point. Line 166 – 167 "ensure that the training data were not biased" – feature scaling does not ensure unbiased datasets, it simply helps the model learn better. The data could still be biased. Line 173: "undergoes feature scaling through the backpropagation algorithm" – not clear what is meant by feature scaling in this context. Line 128 "16 meteorological forecast variables were created by the WRF model" – I believe what is meant here is that 16 variables were extracted as features, but many more variables were generated by the WRF model.

125

**Question 4-1.** Line 161 "average weather and asir quality prediction data". What is meant by average here? Spatial or temporal. Are WRF and CMAQ data extracted from the entire Seoul area domain or subset corresponding to the location of the observation point.

**Answer 4-1.** In "average weather and air quality prediction data" - "average" refers to conversion of 1-h interval data into 6-h
130 interval data. In addition, spatially, it means the average of 9 km grids corresponding to Seoul. We have clarified the meaning and revised it in the paper.

**Question 4-2.** Line 166 – 167 "ensure that the training data were not biased" – feature scaling does not ensure unbiased datasets, it simply helps the model learn better. The data could still be biased.

135 **Answer 4-2.** We thank the reviewer for highlighting this issue. We will incorporate changes based on the suggestion of the reviewer.

Original: Feature scaling, involving standardization and normalization, was used to convert the data into a uniform format, ensure that the training data were not biased and that equal learning took place for the DNN model in each T-step.

Revise: The feature scaling, including standardization and normalization, was implemented to transform data into uniform
140 formats, reduce data bias of training data, and ensure equal learning for the DNN model at each T-step.

**Question 4-3.** Line 173: "undergoes feature scaling through the backpropagation algorithm" – not clear what is meant by feature scaling in this context.

**Answer 4-3.** The phrase "undergoes feature scaling through the backpropagation algorithm" means that feature scaling data
145 is used as training data for the DNN model. We will clarify the meaning in the revised manuscript.

**Question 4-4.** Line 128 "16 meteorological forecast variables were created by the WRF model" – I believe what is meant here is that 16 variables were extracted as features but many more variables were generated by the WRF model.

**Answer 4-4.** "16 meteorological forecast variables were created by the WRF model" - In the paper, the reason for using the
150 weather forecast data was explained through several reference papers in section 2.1. Additionally, $PM_{2.5}$ is discharged from the ground, and it moves at an altitude of 1.5 km or less. Therefore, in this paper, lower altitude data were used. We will add this content in the revised manuscript.

**Question 4-5.** The manuscript could be improved to enhance readability and replicability of the study. I appreciate the authors
155 making code and data available on Zenodo. I would however encourage them to create a GitHub repository with some documentation to allow people easily replicate the results. As mentioned in 1) authors could be more descriptive when detailing data sources.

**Answer 4-5.** As suggested by the reviewer, we upload the code to GitHub. (https://github.com/GercLJB/GMD)

160

**Question 5.** I really don't see the relevance of section 4.2. The models have already been compared and evaluated in terms of predictive skill in regression. Then you take the same models and evaluate in terms of a classification model but only whether

they predicted within those bounds (i.e. the model and results are the same the only thing that changes are the interpretation)

**Answer 5.** In Korea, the PM2.5 forecast results are categorized and provided to the public as good ($PM_{2.5} \leq 15 \ \mu gm^{-3}$), moderate (16 $\mu gm^{-3} \leq PM_{2.5} \leq 35 \ \mu gm^{-3}$), bad (36 $\mu gm^{-3} \leq PM_{2.5} \leq 75 \ \mu gm^{-3}$), and very bad (76 $\mu gm^{-3} \leq PM_{2.5}$) Therefore, both the statistical and category evaluations are necessary to determine whether the DNN model developed in this paper is suitable for forecasting. Section 4.2 presents the comparison of the category performance of the DNN-ALL model and that of the CMAQ model to identify the superior model for actual prediction.

**Section 2. Minor comments.**

**Question 1.** What is the membership function defined in line 144? Is this the generation of temporal features described in subsequent lines? Why was time data encoded in this manner? It seems more standard to represent as integer values or to convert those integer values to cyclic features (i.e. so that month 12 and month 1 are close to each other rather than far away). I haven't seen this approach used previously and would like to understand the motivation and/or justification.

**Answer 1.** As explained in A3-1 among the answers to Q3 (Section 1), the data was expressed stochastically through the membership function to reflect the characteristics of the monthly change.

**Question 2.** The test period is quite short – 3 months out of 51 months. Was there a reason for this?

**Answer 2.** The data from 2016 to 2018 were used as training data, and those from 2019 were used as evaluation data. The data from January to March 2021 were used as test data to find out the performance when the actual DNN model was predicted.

**Question 3.** Line 159 – 163: This is a quite confusing way to present forecast horizons. I'd suggest to just use hours and present forecast horizons as T06, T12, T18, T24, ... Mixing days + hours and having different chunks within each day is confusing to the reader.

**Answer 3.** The T-step presented in Table 3 of the paper was revised and is shown in Table 7.

Table 7. Statistical performance of the DNN-ALL and Random Forest models.

| Day | T-Step | Time | Composition of learning data |
|-----|--------|------|------------------------------|
| D+0 | T12 | 07:00 to 12:00 | |
| | T18 | 13:00 to 18:00 | |
| | T24 | 19:00 to 00:00 | |
| D+1 | T06 | 01:00 to 06:00 | |
| | T12 | 07:00 to 12:00 | 01~06'o clock Observation data of D+0 |
| | T18 | 13:00 to 18:00 | + |
| | T24 | 19:00 to 00:00 | Forecast data of Tx(x : 06, 12, 19, 24) from CMAQ and WRF |
| D+2 | T06 | 01:00 to 06:00 | |
| | T12 | 07:00 to 12:00 | |
| | T18 | 13:00 to 18:00 | |
| | T24 | 19:00 to 00:00 | |

**Question 4.** Line 167 – 168: I don't quite understand why data was both standardized and normalized? Did this improve performance versus just using normalization (if you wished to have bounded between 0 and 1) or indeed versus the unscaled data? Generally people chose either standardization or normalization so I'm curious why you did both.

**Answer 4.** The normal distribution of input variables was standardized through standardization. The normalization was applied thereafter to ensure that the scale of each variable is equal. The reason why both normalization and standardization were applied was to train the characteristics of input variables equally to the DNN model.

**Question 5.** Figure 4: What does Epoch_n = Epoch_n-1 + 1 mean? What does Epoch_n-1 of validation cost > Epoch_n of

validation cost mean? Should it be Validation cost of Epoch_n?

**Answer 5.** While addressing the concern raised by the reviewer, we found out that the formula was incorrect. The modified picture is shown in Fig 1. First, "$Epoch_n=Epoch_{n-1}+1$" expresses that the epoch increases by one as the algorithm is repeated. We modified this part to "$Epoch_{n+1} = Epoch_n + 1$". In addition, "$Epoch_{n-1}$ of Validation cost > $Epoch_n$ of Validation cost" is an incorrect expression, and we will revise it as "Validation cost of $Epoch_{n-1}$ > Validation cost of $Epoch_n$". In addition, in this part, we found that the inequality sign was incorrectly marked, and it was corrected to "Validation cost of $Epoch_{n-1}$ < Validation cost of $Epoch_n$".

Figure 2. Structure of DNN model training process.

**Question 6.** Equation 11 and 12, I'm not sure the use of both MSE and RMSE is necessary and could probably drop one.

**Answer 6.** In this paper, the MSE was used as the cost function of the DNN model. Therefore, it was intended to indicate the degree of difference in MSE for each model. In addition, the RMSE was presented to compare the model errors in units of the PM$_{2.5}$ concentration.

**Question 7.** Line 225 – 230: In classification problems, accuracy, precision and recall are the standard metrics presented. I would suggest in this paper you also include (you already do accuracy so I suggest adding precision and recall)

**Answer 7.** Based on the suggestion of the reviewer, we have added the precision and recall in Table 8. The precision and recall of all categories for the DNN-ALL model in D+0 is presented to be 0.12 equally higher than that of the CMAQ model. However, in D+1 and D+2, the precision of the DNN-ALL and the CMAQ models is found to be similar, and the recall of the DNN-ALL model shows a slight decrease compared to the CMAQ model. The reason for these results is that the two indexes of the good and moderate categories of DNN-ALL are reduced compared to the CMAQ model.

On the other hand, the precision and recall of the DNN-ALL model in the bad category are presented to be higher than those of the CMAQ model. In the bad category of D+0, the precision and recall of DNN-ALL are greater than those of the CMAQ model by 0.24 and 0.04, respectively. In addition, in the very bad category, the precision and recall of DNN-ALL are to be 1.0 equally higher than those of the CMAQ model. In D+1, the precision of DNN-ALL in the bad category is greater than that of the CMAQ model by 0.1, but the recall is similar to the CMAQ model. In D+2, the precision and recall for the bad category of DNN-ALL increased by 0.14 and 0.20 compared to the CMAQ model, respectively. These results show that the performance of the DNN-ALL model is superior to that of the CMAQ model for predicting high concentrations that affect the health of the people.

Table 8. Precision and recall of DNN-ALL and CMAQ models by four categories : "good" (PM2.5 ≤ 15 µgm-3), "moderate" (16 µgm-3 ≤PM2.5 ≤35 µgm-3), "bad" (36 µgm-3 ≤ PM2.5 ≤ 75 µgm-3), and "very bad" (76 µgm-3 ≤ PM2.5).

| Model | Day | Precision | | | | | Recall | | | | |
|---|---|---|---|---|---|---|---|---|---|---|---|
| | | Good | Moderate | Bad | Very bad | Total | Good | Moderate | Bad | Very bad | Total |
| DNN-ALL | D+0 | 0.83 | 0.71 | 0.88 | 1.0 | 0.86 | 0.70 | 0.85 | 0.71 | 1.0 | 0.82 |
| | D+1 | 0.83 | 0.61 | 0.64 | 0.0 | 0.52 | 0.38 | 0.79 | 0.70 | 0.0 | 0.47 |
| | D+2 | 0.79 | 0.59 | 0.56 | 0.0 | 0.49 | 0.42 | 0.67 | 0.75 | 0.0 | 0.46 |
| CMAQ | D+0 | 0.74 | 0.67 | 0.64 | 0.0 | 0.51 | 0.64 | 0.68 | 0.67 | 0.0 | 0.50 |
| | D+1 | 0.85 | 0.69 | 0.54 | 0.0 | 0.52 | 0.65 | 0.67 | 0.70 | 0.0 | 0.51 |
| | D+2 | 0.82 | 0.69 | 0.42 | 0.0 | 0.48 | 0.69 | 0.63 | 0.55 | 0.0 | 0.47 |

**Question 8.** Line 280 – 281: This is a difficult narrative to support. You are using the CMAQ model output as the training data and then saying the usage of that training data allows the model to better represent the long-term-transport-induced phenomenon.

**Answer 8.** In the DNN model (DNN-OBS) that uses only observation data, the RMSE for three months is 11.4 $\mu gm^{-3}$ in D+0. The predictive performance is improved by decreasing the RMSE to 9.4 $\mu gm^{-3}$ for three months, excluding high concentration cases (February 11-14, 2021) by long-distance transport. These results imply that the prediction of high concentration occurrence due to long-distance transportation is insufficient in the case of the DNN model when only the observation data are used. On the other hand, when both the observation data and prediction data are used in the DNN model (DNN-ALL), the RMSE for three months is 7.3 $\mu gm^{-3}$, and the RMSE for excluding high concentration cases by long-distance transport is 7.0 $\mu gm^{-3}$, showing no significant difference. In addition, the RMSE of DNN-ALL is reduced compared to the CMAQ model, showing a superior performance. Therefore, it is inferred that the use of prediction data produced by CMAQ improved the predictive performance of high concentration phenomena by long-distance transport.

**Question 9.** What do the dashed lines in the residual figures represent in Figure 7.

**Answer 9.** In Fig. 7 (a1), (b1), and (c1), each of the dashed lines represents values of 15.5 $\mu gm^{-3}$, 35.5 $\mu gm^{-3}$, 75.5 $\mu gm^{-3}$, and average value of observation (27 $\mu gm^{-3}$). Moreover, 15.5 $\mu gm^{-3}$ is the boundary between "Good" and "Moderate," 35.5 $\mu gm^{-3}$ is the boundary between "Moderate" and "Bad," and 75.5 $\mu gm^{-3}$ is the boundary between "Bad" and "Very bad."

In Fig. 7 (a2), (b2) and (c2), the dashed lines represent the standard deviation of observed $PM_{2.5}$ as negative and positive.

---

## Author Comment (AC2)

We thank the editor and the reviewers for the time and effort put in towards the review of this manuscript. The insightful comments and suggestions have helped improve the manuscript significantly. We have incorporated several changes based on the suggestions of the reviewers. The detailed responses to the reviewers' comments are given below. In Section 1, the answers to major comments are provided, and in Section 2, answers to minor comments are given.

5

**Section 1. Major comments.**

**Q1.** Line 75~80: "In addition, the membership function was used to reflect temporal information." More information is needed about "membership function". How does this function reflect temporal information?

A1. The concept of the membership function is derived from the fuzzy theory, and it defines the probability that a single element belongs to a set. In this study, the probability that the date (element) belongs to 12 months (set) was calculated using the membership function. The date change probability was trained as a factor that reflected the characteristics of the monthly change. As shown in Figure 1 (Figure 5 in the paper), the PM2.5 concentration in Seoul is high in January, February, March, and December, and low from August to October. PM2.5 concentration has a characteristic that changes gradually from month to month. In this paper, the membership function was used to reflect these monthly change characteristics. The examples of how membership function is applied are described in lines 151–153 of the paper.

Figure 1. Time series of the monthly average PM2.5 concentrations from 2016 to 2019.

20

**Q2.** Line 145-155: The authors want to predict PM2.5 within 3 days. Why do you need to add the time information ("adjacent month") of the next month that hasn't happened yet in Eq. (2)? If you know the information of next month, you can predict PM2.5 of next month. Why only forecast PM2.5 within 3 days. This is very difficult to understand.

25

A2. As described in A1, the probability of reflecting the characteristics of the monthly change was calculated using the membership function. The calculated probability was referred to as "adjacent month" and "month." Therefore, "adjacent month" is not a factor that provides information for the next month, but the one that represents the characteristics of the monthly change according to the corresponding date.

**Q3.** In Section 2.2, what are the super parameters of DNN model? Why only use five stacked-layer DNN model? Generally, a neural network model with more than 8 hidden layers is considered as a deep neural network (Hinton et al., 2012).

A3. The statistical and AQI evaluation results of the DNN-ALL model based on the layer are presented in Tables 4 and 5, respectively. The results of the 4-layer and 5-layer models indicate similar performance. However, compared to the 4-layer model, the RMSE of the 5-layer model decreases by approximately 0.1  $\mu$ gm-3 to 1  $\mu$ gm-3 at D+0 to D+2, and the ACC of the 5-layer model increases by approximately 1 %p to 6 %p at D+0 to D+2. Therefore, the 5-layer model shows a superior

35 performance.

30

The 6-layer and 8-layer models have a problem of errors that converge without any decrease in the training process of the model (vanishing gradient problem). The authors believe that the reason for this problem is the activate function. Therefore, as the layer becomes deeper, the value of the last output cannot be significantly changed due to the sigmoid function.

| Model     | Day | MSE
((µgm -3 ) 2 ) | RMSE
(µgm -3 ) | R    | IOA  |
|-----------|-----|---------------------------------------------|------------------------------|------|------|
|           | D+0 | 59.3                                        | 7.7                          | 0.91 | 0.94 |
| 2-layer   | D+1 | 92.1                                        | 9.6                          | 0.86 | 0.89 |
|           | D+2 | 156.3                                       | 12.5                         | 0.75 | 0.80 |
|           | D+0 | 54.7                                        | 7.4                          | 0.91 | 0.95 |
| 4-layer   | D+1 | 88.3                                        | 9.4                          | 0.86 | 0.90 |
| _         | D+2 | 134.5                                       | 11.6                         | 0.77 | 0.84 |
| 5-layer   | D+0 | 53.3                                        | 7.3                          | 0.91 | 0.95 |
| (DNN-ALL) | D+1 | 81.0                                        | 9.0                          | 0.85 | 0.90 |
|           | D+2 | 112.4                                       | 10.6                         | 0.79 | 0.86 |
|           | D+0 | 174.2                                       | 13.2                         | 0.81 | 0.66 |
| 6-layer   | D+1 | 292.4                                       | 17.1                         | 0    | 0.17 |
|           | D+2 | 292.4                                       | 17.1                         | 0    | 0.17 |
|           | D+0 | 302.7                                       | 17.4                         | 0    | 0.15 |
| 8-layer   | D+1 | 292.4                                       | 17.1                         | 0    | 0.17 |
|           | D+2 | 292.4                                       | 17.1                         | 0    | 0.17 |

40 Table 1. Statistical evaluation results according to the number of layers.

Table 2. AQI evaluation results according to the number of layers.

| Model   | Day | ACC  | C (%) | POE  | ) (%) | FAR  | . (%) | F1-score (%) |
|---------|-----|------|-------|------|--------------|------|-------|--------------|
|         | D+0 | 70.0 | 63/90 | 81.8 | 18/22        | 28.0 | 7/25  | 77           |
| 2-layer | D+1 | 55.6 | 50/90 | 81.0 | 17/21        | 39.3 | 11/28 | 69           |
|         | D+2 | 51.1 | 46/90 | 81.0 | 17/21        | 50.0 | 17/34 | 61           |
|         | D+0 | 71.1 | 64/90 | 81.8 | 18/22        | 28.0 | 7/25  | 76           |
| 4-layer | D+1 | 60.0 | 54/90 | 85.7 | 18/21        | 35.7 | 10/28 | 73           |
|         | D+2 | 60.0 | 54/90 | 81.0 | 17/21        | 45.2 | 14/31 | 65           |

| 5-laver     | D+0 | 77.8 | 70/90 | 72.7 | 16/22 | 11.1 | 2/18  | 80 |  |
|-------------|-----|------|-------|------|-------|------|-------|----|--|
| (DNN-ALL)   | D+1 | 64.4 | 58/90 | 71.4 | 15/21 | 31.8 | 7/22  | 70 |  |
| (21(1(1122) | D+2 | 61.1 | 55/90 | 76.2 | 16/21 | 40.7 | 11/27 | 67 |  |
|             | D+0 | 55.6 | 50/90 | 50   | 11/22 | 8.3  | 1/12  | 64 |  |
| 6-layer     | D+1 | 47.8 | 43/90 | 0    | 0/21  | 0    | 0/0   | 0  |  |
|             | D+2 | 47.8 | 43/90 | 0    | 0/21  | 0    | 0/0   | 0  |  |
|             | D+0 | 45.6 | 41/90 | 0    | 0/22  | 0    | 0/0   | 0  |  |
| 8-layer     | D+1 | 47.8 | 43/90 | 0    | 0/21  | 0    | 0/0   | 0  |  |
|             | D+2 | 47.8 | 43/90 | 0    | 0/21  | 0    | 0/0   | 0  |  |
|             |     |      |       |      |       |      |       |    |  |

**Q4.** Line 210~215: The input data of the three experiments (DNN-OBS, DNN-OPM and DNN-ALL) are not very clear. Why should the predicted PM2 5 into the model (DNN-ALL)? Reason?**

A4. The measurement variables presented in Table 1 in Section 2.1 of the paper were used as common variables in the three experiments (DNN-OBS, DNN-OPM, and DNN-ALL). The DNN-OBS used the observation data as the sole training data. Among the predictors shown in Table 2 in Section 2.1 of the paper, the variables produced in the WRF model were used in

50

55

60

Among the predictors shown in Table 2 in Section 2.1 of the paper, the variables produced in the WRF model were used in the DNN-OPM and DNN-ALL experiments, whereas the variables produced in the CMAQ model were used only in the DNN-ALL experiments.

The predicted PM2.5 from CMAQ tends to be over-simulated than the observed PM2.5, but the correlation appears to be good. Therefore, it was judged as training data that can reflect the features of observed PM2.5. The predicted PM2.5, the predicted weather data from WRF, and observation data were studied together to improve PM2.5 prediction performance using DNN-ALL.

Q5. Line 230~240: There's something wrong with Eq. (18). The commonly used expression for F1-score is (2\*ACC\*Recall)/(ACC+Recall). F1-score is for one category. My understanding is that there are four categories (Good, Moderate, Bad and Very bad). Has anyone else used it like this? More explanation is needed.

- A5. The authors agree that F1-score is generally referred to as (2\*Precision\*Recall)/(Precision + Recall). The F1-score used in this paper is not an evaluation of four categories, but an index that simultaneously considers (1-FAR) and POD to evaluate the prediction performance for exceeding 35 µgm-3 as a bad criterion. Tables 3 and 4 show the intervals corresponding to the four and two categories for POD and FAR calculation, respectively. The I in Table 4 is a corresponding category for the
- 65 conditions of a1, a2, b1, and b2 of Table 3. Similarly, II in Table 4 correspond to c1, c2, d1, and d2, III in Table 4 correspond to a3, a4, b3, b4, and IV in Table 4 correspond to c3, c4, d3, and d4. Eq. (7) represents a ratio when prediction concentration in the model corresponds to the observation category in the case that the observation concentration appears in the bad or very bad category. Eq. (8) is the ratio when observation concentration is in the good or moderate category in the case that the prediction concentration appears in bad or very bad category. The POD means Recall, and FAR means (1-precision). Therefore, we was E1 accent to reflect the hormonious mean of POD and (1 EAP).
- 70 we use F1-score to reflect the harmonious mean of POD and (1-FAR).

$$POD(\%) = \frac{(c_3 + c_4 + d_3 + d_4)}{(a_3 + a_4 + b_3 + b_4 + c_3 + c_4 + d_3 + d_4)} \times 100 , \qquad (\underline{15})$$

FAR (%) =
$$\frac{(c1+c2+d1+d2)}{(c1+c2+c3+c4+d1+d2+d3+d4)} \times 100$$
, (26)

Table 3. Intervals corresponding to the four categories for calculating POD and FAR: "good" ( $PM_{2.5} \le 15 \ \mu gm^{-3}$ ), "moderate" (16  $\mu gm^{-3}$ )  $\leq$ PM2.5  $\leq$ 35 µgm-3), "bad" (36 µgm-3  $\leq$  PM2.5  $\leq$  75 µgm-3), and "very bad" (76 µgm-3  $\leq$  PM2.5).

| T                              | 1        |      | Model fo | precast      |             |
|--------------------------------|----------|------|----------|--------------|-------------|
| Level -                        |          | Good | Moderate | Bad          | Very bad    |
|                                | Good     | al   | b1       | c1           | d1          |
| Observation -                  | Moderate | a2   | b2       | c2           | d2          |
| Observation                    | Bad      | a3   | b3       | c3           | d3          |
| -                              | Very bad | a4   | b4       | c4           | d4          |
| $POD = \frac{IV}{II + IV}$ ,   |          |      |          | ( 3 7 | [ -) |
| $FAR = 1 - \frac{IV}{II + IV}$ | ,        |      |          | ( 4 8 | })          |

Table 4. Intervals corresponding to the two categories for calculating POD and FAR : "good and moderate" (PM2.5  $\leq$  35 µgm-3), "bad and very bad" (PM2.5  $\geq$  36 µgm-3).

|             |                      | Model             | forecast         |  |  |
|-------------|----------------------|-------------------|------------------|--|--|
| Level       |                      | Good and moderate | Bad and very bad |  |  |
| Observation | Good and
Moderate | I                 | П                |  |  |
| Observation | Bad and
Very bad  | Ш                 | IV               |  |  |

80

90

75

Q6. In Table 2, why are 925hPa and 850hPa variables selected? Why not consider 700hPa and 500hp variables? Is there any reason?

A6. Various forecast data for each altitude are produced in the WRF model. However, the reason why the upper layer altitude 85 (700 and 500 hPa) was not used in this study is that the emission of PM2.5 mainly occurs on the ground and moves up to an altitude of 1.5 km. Therefore, we only used the lower altitude forecast data.

Q7. Table 5 only provides the performance of the model in the test set (January–March 2021) and it is recommended to add the performance of the model in the training set (2016 to 2018) and validation set (2019).

A7. Table 3 shows the statistical evaluation results of three experiments (DNN-OBS, DNN-OPM, and DNN-ALL) and CMAQ models from 2016 to 2018. In D+0 to D+2, the DNN-ALL model performs the best in terms of all statistical indicators. In

addition, the values of all three experiments indicate a decrease in the RMSE compared to the CMAQ model.

95

Table 4 presents the statistical evaluation results of the three experiments (DNN-OBS, DNN-OPM, and DNN-ALL) and CMAQ models for 2019. The DNN-OBS model shows similar performance for D+0 compared to the CMAQ model but decreased performance owing to an increased RMSE of D+1 and D+2 by 2 µgm-3 and 2.2 µgm-3, respectively. The DNN-OPM model shows an increase in performance owing to a decrease in the RMSE of D+0 by 3 µgm-3 compared to the CMAQ model. Moreover, the RMSE of D+1 and D+2 decrease by 0.4 µgm-3 and 0.4 µgm-3 compared to the CMAQ model, respectively, indicating that the performance is similar. For the DNN-ALL model, the RMSE from D+0 to D+2 100 decreased by 4.6 µgm-3, 2.7 µgm-3, and 2.1 µgm-3, compared to the CMAQ model, which shows an improved performance.

Table 53. Statistical evaluation results of CMAQ, DNN-OBS, DNN-OPM, and DNN-ALL models from 2016 to 2018.

| Model   | Day $\frac{MSE}{((\mu gm^{-3})^2)}$ |       | RMSE
(µgm -3 ) | R    | IOA  |
|---------|-------------------------------------|-------|------------------------------|------|------|
|         | D+0                                 | 136.9 | 11.7                         | 0.76 | 0.86 |
| CMAQ    | D+1                                 | 146.4 | 12.1                         | 0.74 | 0.84 |
|         | D+2                                 | 185.0 | 13.6                         | 0.67 | 0.80 |
|         | D+0                                 | 79.2  | 8.9                          | 0.78 | 0.87 |
| DNN-OBS | D+1                                 | 139.2 | 11.8                         | 0.54 | 0.65 |
| _       | D+2                                 | 158.8 | 12.6                         | 0.43 | 0.54 |
|         | D+0                                 | 53.3  | 7.3                          | 0.86 | 0.92 |
| DNN-OPM | D+1                                 | 88.4  | 9.4                          | 0.75 | 0.83 |
|         | D+2                                 | 108.2 | 10.4                         | 0.68 | 0.77 |
|         | D+0                                 | 39.7  | 6.3                          | 0.90 | 0.94 |
| DNN-ALL | D+1                                 | 57.8  | 7.6                          | 0.84 | 0.90 |
|         | D+2                                 | 72.3  | 8.5                          | 0.80 | 0.87 |

105

Table 64. Statistical evaluation results of CMAQ, DNN-OBS, DNN-OPM, and DNN-ALL models for 2019.

| Model   | Day | MSE
((µgm -3 ) 2 ) | RMSE
(µgm -3 ) | R    | ΙΟΑ  |
|---------|-----|---------------------------------------------|------------------------------|------|------|
|         | D+0 | 123.2                                       | 11.1                         | 0.82 | 0.90 |
| CMAQ    | D+1 | 132.3                                       | 11.5                         | 0.80 | 0.89 |
|         | D+2 | 156.3                                       | 12.5                         | 0.75 | 0.86 |
|         | D+0 | 92.2                                        | 9.6                          | 0.84 | 0.88 |
| DNN-OBS | D+1 | 182.3                                       | 13.5                         | 0.63 | 0.65 |
|         | D+2 | 216.1                                       | 14.7                         | 0.52 | 0.52 |
|         | D+0 | 65.6                                        | 8.1                          | 0.89 | 0.92 |
| DNN-OPM | D+1 | 123.2                                       | 11.1                         | 0.78 | 0.81 |
|         | D+2 | 166.4                                       | 12.9                         | 0.66 | 0.72 |

| DNN-ALL | D+0 | 42.3  | 6.5  | 0.93 | 0.95 |
|---------|-----|-------|------|------|------|
|         | D+1 | 77.4  | 8.8  | 0.88 | 0.90 |
|         | D+2 | 108.2 | 10.4 | 0.81 | 0.84 |

**Q8.** In Table 5: The DNN-ALL model uses the forecast variable (F\_PM2.5 predicted by CMAQ). However, IOA of F\_PM2.5 in CMAQ is 0.9, 0.9 and 0.85 respectively, and IOA of PM2.5 in DNN-ALL is 0.95, 0.9 and 0.86 respectively. Could it be

110 understood that compared with CMAQ, IOA in DNN-ALL model is improved by 0.05, 0.0 and 0.01 respectively? More explanation is needed.

**A8.** The denominator of IOA indicates the trends of the model and observation based on the average of observation, and the numerator of IOA represents the deviation of the model and observation. In other words, the IOA can be interpreted as an indicator that considers trends and quantitative differences. Therefore, the quantitative difference (error) of the DNN-ALL

115 model decreases compared to the CMAQ model, but the trend toward the mean of observation is similar between the two models, showing no significant difference in IOA.

**Q9.** In Table 6: From T04 to T11, why does the indicators (RMSE and IOA) not decrease monotonically? The IOA of T09 is larger than T04. Meanwhile, the mean IOA of D+2 is 0.79 ((0.77+0.85+0.74+0.80)/4.0) and IOA of D+2 in table 5 is 0.86, What are the reasons for the unequal values?

120

**A9.** There could be a difference in the performance of the model according to the conditions of target time such as daytime, nighttime, high concentration, and low concentration. As shown in the CMAQ results, the prediction performance of the DNN-ALL model degrades or improves monotonically over time.

The IOA of D+2 is not equal to (0.77+0.85+0.74+0.80)/4.0. The IOA of D+2 refers to the value calculated using the IOA method after calculating the daily average concentration using the predicted concentration of each T-step such as T08, T09, T10, and T11. (0.77+0.85+0.74+0.80)/4.0 is simply averaged after calculating IOA using the predicted concentration by T-step.

**Section 2. Minor comments.**

130 **Q1.** Line 19: "IOA" should be "index of agreement (IOA)". The first abbreviation needs to give the complete name. Please check other parts of the paper.

**A1.** We thank the reviewer for highlighting this issue. We have included the complete name at the first mention of the abbreviation.

**135**

Q2. Line 100 Figure 3: Nested-grid is often used in models. It is recommended to combine figure 2 and figure 3 into one figure.A2. Figure 2 shows the location information of the measuring station where the measurement data are obtained, and Figure 3 depicts the domain of the model. Therefore, the information conveyed by the two images is different.

140

145

**Q3. Add the temporal and spatial resolution of the variables in Tables 1 and 2.**

A3. Based on the suggestion of the reviewer, the descriptions are added to Table 5 and Table 6.

Table 75. Training variables in the PM2.5 prediction system using a DNN based on surface-weather observations. Air quality variables are obtained from 41 air quality measurement equipment in Seoul. Surface weather variables are obtained from ASOS in Seoul. Observation data are collected every hour.

| Observation
Variable | Description                                                  | Unit                             |
|-------------------------|--------------------------------------------------------------|----------------------------------|
| O_SO 2       | Sulfur dioxide                                               | ppm                              |
| O_NO 2       | Nitrogen dioxide                                             | ppm                              |
| O_O 3        | Ozone                                                        | ppm                              |
| 0_C0                    | Carbon monoxide                                              | ppm                              |
| O_PM 10      | Particulate matter (aerodynamic diameters $\leq 10 \ \mu$ m) | µgm -3                |
| O_PM 2.5     | Particulate matter (aerodynamic diameters $\leq$ 2.5 µm)     | µgm -3                |
| O_V                     | Vertical wind velocity                                       | m/s                              |
| O_U                     | Horizontal wind velocity                                     | m/s                              |
| O_RN_ACC                | Accumulative precipitation                                   | Mm                               |
| O_RH                    | Relative humidity                                            | %                                |
| O_Td                    | Dew point temperature                                        | °C                               |
| O_Pa                    | Pressure                                                     | hPa                              |
| O_Radiation             | Solar radiation                                              | 0.01 MJ per
hr-m 3 |
| O_Ta                    | Air temperature                                              | °C                               |

Table 36. Training variables in the PM2.5 prediction system using a DNN based on the WRF and CMAQ models. WRF and

| Model | Forecast
Variable     | Description                                                     | Unit              |
|-------|--------------------------|-----------------------------------------------------------------|-------------------|
| CMAQ  | F_PM 2.5      | Particulate matter (aerodynamic diameter $\leq$ 2.5 µm)         | µgm -3 |
|       | F_V                      | Vertical wind velocity at surface                               | m/s               |
| -     | F_U                      | Horizontal wind velocity at surface                             | m/s               |
|       | F_RN_ACC                 | Accumulative precipitation                                      | mm                |
| -     | F_RH                     | Relative humidity at surface                                    | %                 |
| -     | F_Pa                     | Pressure at surface                                             | pa                |
| -     | F_Ta                     | Air temperature at surface                                      | K                 |
|       | F_MH                     | Mixing height                                                   | m                 |
| -     | F_925hpa_gpm             | Position altitude at 925 hPa                                    | m                 |
| WRF   | F_925hpa_V               | Vertical wind velocity at 925 hPa                               | m/s               |
| -     | F_925hpa_U               | Horizontal wind velocity at 925 hPa                             | m/s               |
|       | F_850hpa_gpm             | Position altitude at 850 hPa                                    | m                 |
|       | F_850hpa_V               | Vertical wind velocity at 850 hPa                               | m/s               |
| -     | F_850hpa_U               | Horizontal wind velocity at 850 hPa                             | m/s               |
| -     | F_850hpa_RH              | Relative humidity at 850 hPa                                    | %                 |
| -     | F_850hpa_Ta              | Potential temperature at 850 hPa                                | Θ                 |
| -     | F_Temp_
850hpa-925hpa | Potential temperature difference between
850 hPa and 925 hPa | Θ                 |

CMAQ model results are obtained from 9 km horizontal grid resolution. These values are collected on an hourly interval.

**Q4.** The font in Figure 4 is too small to see clearly.**

155 **A4.** We have increased the font size to improve the clarity of the figure.

---

## Author Response (AR1)

**Author Response to Referee 1**

We thank the editor and the reviewers for the time and effort put in towards the review of this manuscript. The insightful comments and suggestions have helped improve the manuscript significantly. We have incorporated several changes based on the suggestions of the reviewers. The detailed responses to the reviewers' comments are given below.

**Section 1. Major comments.**

**Comment 1.** It's not clear what the contributions of the paper are versus what already exists. Did the authors run the WRF and CMAQ models to generate the training data or were these data obtained from some other source?

**Response 1.** We understand the concern raised by the reviewer. We directly generated the training data using the WRF and CMAQ. We will make revisions to the manuscript to clarify this point. (See lines 128-129 in the revised manuscript)

10

15

20

**Comment 2.** Similarly there is no justification for the choice of model. I would like to see the approach benchmarked against simpler models (ARIMA, Random Forest, ... basically anything from the statistical or machine learning family to compare against the deep learning approach). The volume of data that the model is trained on are not huge so it is not apparent that a DNN is the best choice of algorithm.

**Comment 2-1**. Similarly, there is no justification for the choice of model. I would like to see the approach benchmarked against simpler models (ARIMA, Random Forest, ... basically anything from the statistical or machine learning family to compare against the deep learning approach).

**Response 2-1.** Among the statistical models mentioned by the reviewer, the prediction performances of the Random Forest (RF) have been evaluated and compared with that of the DNN-ALL. Table 1 shows the results of the statistical evaluations, and Table 2 lists the results of the Air Quality Index (AQI) evaluation (**Tables 1 and 2 are reflected in Tables S5 and S7 of Supplement, respectively**).

- 25 Compared to the results of the RF model, the Root Mean Square Error (RMSE) value of the DNN-ALL model decreased by 0.6 to 1.9 µgm-3, and the Correlation Coefficient (R) and Index of Agreement (IOA) values increased slightly (See lines 340-343 in the revised manuscript). The Accuracy (ACC) of the DNN-ALL model increased by approximately 2–13 %p compared to the RF model, and the F1-score decreased by 1 %p at D+1 but increased by 1 %p and 9 %p at D+0 and D+2, respectively (See lines 398-400 in the revised manuscript). A comparison of the performance results showed that the DNN-20 and DE and
- 30 ALL model outperformed the RF model. We will this information in the revised manuscript to include these results.

| Model   | Day | MSE
((µgm -3 ) 2 ) | RMSE
(µgm -3 ) | R    | ΙΟΑ  |
|---------|-----|---------------------------------------------|------------------------------|------|------|
| DNN-ALL | D+0 | 53.3                                        | 7.3                          | 0.91 | 0.95 |
|         | D+1 | 81.0                                        | 9.0                          | 0.85 | 0.90 |
|         | D+2 | 112.4                                       | 10.6                         | 0.79 | 0.86 |

Table 1. Statistical performance of the DNN-ALL and Random Forest models.

| Random Forest | D+0 | 62.4  | 7.9  | 0.90 | 0.93 |
|---------------|-----|-------|------|------|------|
|               | D+1 | 106.1 | 10.3 | 0.83 | 0.85 |
|               | D+2 | 156.3 | 12.5 | 0.73 | 0.76 |

Table 2. Categorical performance of the DNN-ALL and Random Forest model.

| Model            | Day | ACC (%) |       | POE  | POD (%) |      | . (%) | F1-score (%) |
|------------------|-----|---------|-------|------|---------|------|-------|--------------|
| DNN-ALL          | D+0 | 77.8    | 70/90 | 72.7 | 16/22   | 11.1 | 2/18  | 80           |
|                  | D+1 | 64.4    | 58/90 | 71.4 | 15/21   | 31.8 | 7/22  | 70           |
|                  | D+2 | 61.1    | 55/90 | 76.2 | 16/21   | 40.7 | 11/27 | 67           |
| Random
Forest | D+0 | 75.6    | 68/90 | 77.3 | 17/22   | 19.0 | 4/21  | 79           |
|                  | D+1 | 61.1    | 55/90 | 76.2 | 16/21   | 33.3 | 8/24  | 71           |
|                  | D+2 | 48.9    | 44/90 | 71.4 | 15/21   | 50.0 | 15/30 | 58           |

**Comment 2-2**. The volume of data that the model is trained on are not huge so it is not apparent that a DNN is the best choice of algorithm.

**Response 2-2.** We agree that the volume of data in this paper is not sufficiently huge to be applied to artificial intelligence (AI). Nevertheless, the reason for choosing DNN algorithm is to take into account the scalability of the model, which can reflect training data expansion to forecast the segmentation with a 1-h interval and the future data growth over time. Therefore, the performance of the AI is expected to improve as the training data increases. (See lines 343-346 in the revised manuscript)

40

55

60

Comment 3. Many details on the DNN model setup are presented with no real justification for their choice e.g. using the
membership function for temporal features, the choice of DNN architecture such as number of layers is not explained. On line 50 – 55 the authors 1) note the advantages of RNN for time series forecasting and 2) that Kim et al. (2019) developed an RNN model to predict PM2.5 concentrations at two locations in Seoul. Why was RNN not considered for this study rather than DNN and how does the performance of this model compare to that reported by Kim et. al. Similarly it is not clear if the autoregressive features of the data were expressed in any form? Of course RNN expresses these implicitly but in other models it can be advantageous to feature engineer the autoregressive dependencies. Were any feature combinations other than those reported explored in the paper. The authors need to justify the choice of algorithm and how the DNN was designed detailing such information as feature selection, number of layers/nodes.

**Comment 3-1.** Many details on the DNN model setup are presented with no real justification for their choice e.g. using the membership function for temporal features.

**Response 3-1.** In this paper, the membership function was used to reflect these monthly change characteristics. As shown in Figure 1 (Figure 5 in the paper),  $PM_{2.5}$  concentration in Seoul is high in January, February, March, and December, and low from August to October.  $PM_{2.5}$  concentration has a characteristic that changes gradually from month to month. The examples of how membership function is applied are described in lines **160–163** of the paper. The membership function was applied based on the results presented by Yu et al. (2019). Yu et al. (2019) performed training that reflected monthly change

2

characteristics to improve the high-concentration  $PM_{10}$  forecast performance. As indicated by the experiment results presented in Table 3, the POD performance of the training model reflecting the characteristics of the monthly change was improved by 25 %p. The information related to this will be added in the paper. (See lines 151-153 in the revised manuscript)

Table 3. Results of artificial intelligence model performance evaluation when using and without the membership function presented in Yu et al. (2019).

| Model                          | Day | ACC (%) | POD (%) | FAR (%) |
|--------------------------------|-----|---------|---------|---------|
| Using Membership
function   | D+1 | 70      | 75      | 48      |
| Without Membership
function | D+1 | 76      | 50      | 33      |

Figure 1. Time series of the average monthly PM2.5 concentrations from 2016 to 2019.

70 Comment 3-2. The choice of DNN architecture such as number of layers is not explained. The authors need to justify the choice of algorithm and how the DNN was designed detailing such information as feature selection, number of layers/nodes. Were any feature combinations other than those reported explored in the paper.

**Response 3-2.** In order to provide the justification for the layer selection mentioned by the reviewer, we presented the evaluation results according to the number of layers. The statistical and AQI evaluation results of the DNN-ALL model based

- on the layer are presented in Tables 4 and 5, respectively (Tables 4 and 5 are reflected in Table S1 and S2 of Supplement, respectively). The results of the 4-layer and 5-layer models show that the performance is similar. However, compared with the 4-layer model, the RMSE of the 5-layer decreases by approximately 0.1 µgm-3 to 1 µgm-3 at D+0 to D+2, and the ACC of the 5-layer model increases by approximately 1 %p to 6 %p at D+0 to D+2. Therefore, the 5-layer model shows the best performance. The 6-layer and 8-layer models contain errors that converge without decreasing during the training process of the model (vanishing gradient problem). The authors believe that the cause of this problem is the activate function. Therefore, as
- 80 the model (vanishing gradient problem). The authors believe that the cause of this problem is the activate function. Therefore, as the layer becomes deeper, the value of the last output cannot be significantly changed due to the sigmoid function. We will include this information in the revised manuscript. (See lines 188-194 in the revised manuscript)

Table 4. Statistical evaluation results according to the number of layers.

| Model     | Day | MSE RMSE
((μgm -3 ) 2 ) (μgm -3 ) |      | R    | ΙΟΑ  |
|-----------|-----|-----------------------------------------------------------------------|------|------|------|
|           | D+0 | 59.3                                                                  | 7.7  | 0.91 | 0.94 |
| 2-layer   | D+1 | 92.1                                                                  | 9.6  | 0.86 | 0.89 |
|           | D+2 | 156.3                                                                 | 12.5 | 0.75 | 0.80 |
| 4-layer   | D+0 | 54.7                                                                  | 7.4  | 0.91 | 0.95 |
|           | D+1 | 88.3                                                                  | 9.4  | 0.86 | 0.90 |
|           | D+2 | 134.5                                                                 | 11.6 | 0.77 | 0.84 |
| 5-layer   | D+0 | 53.3                                                                  | 7.3  | 0.91 | 0.95 |
| (DNN-ALL) | D+1 | 81.0                                                                  | 9.0  | 0.85 | 0.90 |
|           | D+2 | 112.4                                                                 | 10.6 | 0.79 | 0.86 |
|           | D+0 | 174.2                                                                 | 13.2 | 0.81 | 0.66 |
| 6-layer   | D+1 | 292.4                                                                 | 17.1 | 0    | 0.17 |
|           | D+2 | 292.4                                                                 | 17.1 | 0    | 0.17 |
|           | D+0 | 302.7                                                                 | 17.4 | 0    | 0.15 |
| 8-layer   | D+1 | 292.4                                                                 | 17.1 | 0    | 0.17 |
|           | D+2 | 292.4                                                                 | 17.1 | 0    | 0.17 |

Table 5. AQI evaluation results according to the number of layers.

| Model     | Day | ACC  | C (%) | POE  | POD (%) |      | . (%) | F1-score (%) |
|-----------|-----|------|-------|------|---------|------|-------|--------------|
|           | D+0 | 70.0 | 63/90 | 81.8 | 18/22   | 28.0 | 7/25  | 77           |
| 2-layer   | D+1 | 55.6 | 50/90 | 81.0 | 17/21   | 39.3 | 11/28 | 69           |
|           | D+2 | 51.1 | 46/90 | 81.0 | 17/21   | 50.0 | 17/34 | 61           |
|           | D+0 | 71.1 | 64/90 | 81.8 | 18/22   | 28.0 | 7/25  | 76           |
| 4-layer   | D+1 | 60.0 | 54/90 | 85.7 | 18/21   | 35.7 | 10/28 | 73           |
|           | D+2 | 60.0 | 54/90 | 81.0 | 17/21   | 45.2 | 14/31 | 65           |
| 5-laver   | D+0 | 77.8 | 70/90 | 72.7 | 16/22   | 11.1 | 2/18  | 80           |
| (DNN-ALL) | D+1 | 64.4 | 58/90 | 71.4 | 15/21   | 31.8 | 7/22  | 70           |
| · · · · · | D+2 | 61.1 | 55/90 | 76.2 | 16/21   | 40.7 | 11/27 | 67           |
|           | D+0 | 55.6 | 50/90 | 50   | 11/22   | 8.3  | 1/12  | 64           |
| 6-layer   | D+1 | 47.8 | 43/90 | 0    | 0/21    | 0    | 0/0   | 0            |
|           | D+2 | 47.8 | 43/90 | 0    | 0/21    | 0    | 0/0   | 0            |
|           | D+0 | 45.6 | 41/90 | 0    | 0/22    | 0    | 0/0   | 0            |
| 8-layer   | D+1 | 47.8 | 43/90 | 0    | 0/21    | 0    | 0/0   | 0            |
|           | D+2 | 47.8 | 43/90 | 0    | 0/21    | 0    | 0/0   | 0            |

**Comment 3-3.** On line 50 – 55 the authors 1) note the advantages of RNN for time series forecasting. and 2) that Kim et al. (2019) developed an RNN model to predict PM2.5 concentrations at two locations in Seoul. Why was RNN not considered for this study rather than DNN and how does the performance of this model compare to that reported by Kim et. al.

90

**Response 3-3-1. (Reason why RNN was not considered)** There are very few studies and relatively less research to predict air quality using AI such as DNN, RNN and CNN, although it has increased recently. Therefore, the purpose of this study is to evaluate the performance of fine dust prediction when using the DNN among various AI algorithms. The RNN is known to have the advantage of time series prediction, and the DNN is known to have the advantage of extracting characteristics of training data well. There is no convergent result confirming which of the two algorithms is better when applied to fine dust prediction. Therefore, we first performed the simulation using the DNN rather than the RNN in order to maximize the advantages of the DNN for predicting fine dust. In the future, we plan to perform comparative evaluation with the DNN results presented in this paper through the development of RNN models.

- 100 Response 3-3-2. (Comparison with Kim et al.) We compared the results obtained by Kim et al. (2019) with those obtained in our study. Kim et al. (2019) performed a PM2.5 concentration prediction for two out of 41 measuring stations that are located in the Seoul area. However, in this paper, the average PM2.5 concentration prediction for 41 measuring stations in Seoul was performed. In other words, there is a spatial difference for the area to be predicted. In addition, the periods of prediction for the two papers are different. The forecast period considered by Kim et al. (2019) was four months, from January 2016 to April
   105 2016 and the formula to the prediction of the formula to the prediction of the two papers are different. The forecast period considered by Kim et al. (2019) was four months, from January 2016 to April
- 105 2016, and the forecast period in this study was three months, from January 2021 to March 2021. Although it is difficult to directly compare the two studies because of the existence of temporal and spatial differences, the results of the prediction performance are presented in Table 6. Because Kim et al. (2019) performed only the D+1 prediction, the comparison of the prediction performance with this paper was conducted for D+1. The values indicate that the RMSE is decreased and the IOA is increased compared to other models.
- 110

95

| Model                       | Day | RMSE
(µgm -3 ) | ΙΟΑ  |
|-----------------------------|-----|------------------------------|------|
| DNN-ALL                     | D+1 | 9.0                          | 0.90 |
| Seoul-1 (Kim et al. (2019)) | D+1 | 12.5                         | 0.71 |
| Seoul-2 (Kim et al. (2019)) | D+1 | 15.1                         | 0.77 |

Table 6. Statistical performance of the DNN-ALL and Kim et al. (2019).

**Comment 3-4.** Similarly, it is not clear if the autoregressive features of the data were expressed in any form? Of course, RNN expresses these implicitly but in other models it can be advantageous to feature engineer the autoregressive dependencies.

**Response 3-4.** The RNN algorithm implicitly reflects autoregressive features, but the DNN algorithm does not reflect autoregressive features. This study did not consider any autoregressive features.

120

Comment 4. The manuscript could be improved to enhance readability and replicability of the study. I appreciate the authors
making code and data available on Zenodo. I would however encourage them to create a GitHub repository with some documentation to allow people easily replicate the results. As mentioned in 1) authors could be more descriptive when detailing data sources. Some parts of the paper could be explained better, e.g. line 161 "average weather and air quality prediction data". What is meant by average here? Spatial or temporal. Are WRF and CMAQ data extracted from the entire Seoul area domain

or subset corresponding to location of the observation point. Line 166 - 167 "ensure that the training data were not biased" -

- 125 feature scaling does not ensure unbiased datasets, it simply helps the model learn better. The data could still be biased. Line 173: "undergoes feature scaling through the backpropagation algorithm" – not clear what is meant by feature scaling in this context. Line 128 "16 meteorological forecast variables were created by the WRF model" – I believe what is meant here is that 16 variables were extracted as features, but many more variables were generated by the WRF model.
- 130 Comment 4-1. Line 161 "average weather and asir quality prediction data". What is meant by average here? Spatial or temporal. Are WRF and CMAQ data extracted from the entire Seoul area domain or subset corresponding to location of the observation point.

**Response 4-1.** In "average weather and air quality prediction data" - "average" refers to conversion of 1-h interval data into 6-h interval data. In addition, spatially, it means the average of 9 km grids corresponding to Seoul. We have clarified the meaning and revised it in the paper. (See lines 171-174 in the revised manuscript)

**Comment 4-2.** Line 166 - 167 "ensure that the training data were not biased" – feature scaling does not ensure unbiased datasets, it simply helps the model learn better. The data could still be biased.

**Response 4-2.** We thank the reviewer for highlighting this issue. We will incorporate changes based on the suggestion of the reviewer.

Original: Feature scaling, involving standardization and normalization, was used to convert the data into a uniform format, ensure that the training data were not biased and that equal learning took place for the DNN model in each T-step.

Revise: The feature scaling, including standardization and normalization, was implemented to transform data into uniform formats, reduce data bias of training data, and ensure equal learning for the DNN model at each T-step. (See lines 178-179 in the revised menuscript)

145 the revised manuscript)

135

140

**Comment 4-3.** Line 173: "undergoes feature scaling through the backpropagation algorithm" – not clear what is meant by feature scaling in this context.

Response 4-3. The phrase "undergoes feature scaling through the backpropagation algorithm" means that feature scaling data
is used as training data for the DNN model. We will clarify the meaning in the revised manuscript. (See lines 187-188 in the revised manuscript)

**Comment 4-4.** Line 128 "16 meteorological forecast variables were created by the WRF model" – I believe what is meant here is that 16 variables were extracted as features but many more variables were generated by the WRF model.

- 155 **Response 4-4.** "16 meteorological forecast variables were created by the WRF model" In the paper, the reason for using the weather forecast data was explained through several reference papers in section 2.1. Additionally, PM2.5 is discharged from the ground, and it moves at an altitude of 1.5 km or less. Therefore, in this paper, lower altitude data were used. We will add this content in the revised manuscript. (See lines 131-133 in the revised manuscript)
- 160 **Comments 4-5.** The manuscript could be improved to enhance readability and replicability of the study. I appreciate the authors making code and data available on Zenodo. I would however encourage them to create a GitHub repository with some

documentation to allow people easily replicate the results. As mentioned in 1) authors could be more descriptive when detailing data sources.

Response 4-5. As suggested by the reviewer, we upload the code to GitHub. (https://github.com/GercLJB/GMD) (See line 438 in the revised manuscript)

**165**

Comment 5. I really don't see the relevance of section 4.2. The models have already been compared and evaluated in terms of predictive skill in regression. Then you take the same models and evaluate in terms of a classification model but only whether they predicted within those bounds (i.e. the model and results are the same the only thing that changes are the

**interpretation)**

170

195

**Response 5.** In Korea, the PM2.5 forecast results are categorized and provided to the public as good ( $PM_{2.5} \le 15 \ \mu gm^{-3}$ ), moderate (16  $\mu$ gm-3  $\leq$  PM2.5  $\leq$  35  $\mu$ gm-3), bad (36  $\mu$ gm-3  $\leq$  PM2.5  $\leq$  75  $\mu$ gm-3), and very bad (76  $\mu$ gm-3  $\leq$  PM2.5) Therefore, both the statistical and category evaluations are necessary to determine whether the DNN model developed in this paper is suitable for

175 forecasting. Section 4.2 presents the comparison of the category performance of the DNN-ALL model and that of the CMAQ model to identify the superior model for actual prediction.

**Section 2. Minor comments.**

180 Comment 1. What is the membership function defined in line 144? Is this the generation of temporal features described in subsequent lines? Why was time data encoded in this manner? It seems more standard to represent as integer values or to convert those integer values to cyclic features (i.e. so that month 12 and month 1 are close to each other rather than far away). I haven't seen this approach used previously and would like to understand the motivation and/or justification.

Response 1. As explained in Response 3-1 among the answers to Comments 3 (Section 1), the data was expressed 185 stochastically through the membership function to reflect the characteristics of the monthly change.

**Comment 2.** The test period is quite short -3 months out of 51 months. Was there a reason for this?**

Response 2. The data from 2016 to 2018 were used as training data, and those from 2019 were used as evaluation data. The 190 data from January to March 2021 were used as test data to find out the performance when the actual DNN model was predicted.

Comment 3. Line 159 – 163: This is a quite confusing way to present forecast horizons. I'd suggest to just use hours and present forecast horizons as T06, T12, T18, T24, ... Mixing days + hours and having different chunks within each day is confusing to the reader.

Response 3. The T-step presented in Table 3 of the paper was revised and is shown in Table 7 (Table 7 is reflected in Table 3 of the revised manuscript). (See lines 171-172 in the revised manuscript)

| Day   | T-Step             | Time           | Composition of learning data                                                 |  |  |  |  |  |  |
|-------|--------------------|----------------|------------------------------------------------------------------------------|--|--|--|--|--|--|
|       | T 12_D0 | 07:00 to 12:00 |                                                                              |  |  |  |  |  |  |
| D+0   | $T_{18\_D0}$       | 13:00 to 18:00 |                                                                              |  |  |  |  |  |  |
|       | $T_{24}_{D0}$      | 19:00 to 00:00 | _                                                                            |  |  |  |  |  |  |
|       | T 06_D1 | 01:00 to 06:00 |                                                                              |  |  |  |  |  |  |
| D+1 — | $T_{12}D_1$        | 07:00 to 12:00 | 01:00 to 06:00 observations data on D+0 at each T-step
+                  |  |  |  |  |  |  |
|       | $T_{18}D_1$        | 13:00 to 18:00 |                                                                              |  |  |  |  |  |  |
|       | $T_{24\_D1}$       | 19:00 to 00:00 | Forecast data of $T_{x_Dy}$ (x: 06, 12, 18, 24, y: 0 to 2) from CMAQ and WRF |  |  |  |  |  |  |
|       | $T_{06\_D2}$       | 01:00 to 06:00 | _                                                                            |  |  |  |  |  |  |
| D12   | $T_{12\_D2}$       | 07:00 to 12:00 | _                                                                            |  |  |  |  |  |  |
| D+2   | $T_{18\_D2}$       | 13:00 to 18:00 | _                                                                            |  |  |  |  |  |  |
|       | $T_{24\_D2}$       | 19:00 to 00:00 |                                                                              |  |  |  |  |  |  |

Table 7. Configuration of the training data for each T-step to implement the DNN model for the 6-hour average prediction.

**Comment 4.** Line 167 – 168: I don't quite understand why data was both standardized and normalized? Did this improve performance versus just using normalization (if you wished to have bounded between 0 and 1) or indeed versus the unscaled data? Generally people chose either standardization or normalization so I'm curious why you did both.

205 **Response 4.** The normal distribution of input variables was standardized through standardization. The normalization was applied thereafter to ensure that the scale of each variable is equal. The reason why both normalization and standardization were applied was to train the characteristics of input variables equally to the DNN model. (See lines 179-180 and 183-184 in the revised manuscript)

**210**

215

**Comment 5.** Figure 4: What does Epoch\_n = Epoch\_n-1 + 1 mean? What does Epoch\_n-1 of validation cost > Epoch\_n of validation cost mean? Should it be Validation cost of Epoch\_n?

**Response 5.** While addressing the concern raised by the reviewer, we found out that the formula was incorrect. The modified picture is shown in Fig 1. First, "Epochn=Epochn-1+1" expresses that the epoch increases by one as the algorithm is repeated. We modified this part to "Epochn+1 = Epochn + 1". In addition, "Epochn-1 of Validation cost > Epochn of Validation cost" is an incorrect expression, and we will revise it as "Validation cost of Epochn-1> Validation cost of Epochn". In addition, in this part, we found that the inequality sign was incorrectly marked, and it was corrected to "Validation cost of Epochn-1 < Validation cost of Epochn". (Figure 2 is reflected in Figure 4 of the revised manuscript)

---

## Author Response (AR2)

**Author Response to Topic Editor**

We thank the topic editor for the time and effort put in towards the review of this manuscript. The detailed responses to the topic editor comments are given below.

**5    Minor comments.**

**Comment:** Please provide Figure 4 and Figure 7 with larger letters and number and which are not hard to read currently.

**Response:** The resolution of Figure 4 and Figure 7 is increased to provide Figure 4 and Figure 7 with larger letters and number. **(See Figure 4 and Figure 7 of the revised manuscript)**

---

## Author Response (AR3)

**Author Response to Topic Editor**

We thank the topic editor for the time and effort put in towards the review of this manuscript. The detailed comments have helped improve the manuscript significantly. The detailed responses to the topic editor comments are given below.

**Minor comments.**

**Comment.** Could you revise abstract for better transfer of results and implications?

**Response.** We revised the abstract.

**Abstract.** Despite recent progress of numerical air quality models, accurate prediction of fine particulate matter ($PM_{2.5}$) is still challenging because of uncertainties in physical and chemical parameterizations, meteorological data, and emission inventory database. Recent advances in artificial neural network can be used to overcome limitations in numerical air quality models. In this study, a deep neural network (DNN) model was developed for a 3-day forecasting of 6-hour average $PM_{2.5}$ concentrations - the day of prediction (D+0), one day after prediction (D+1) and two days after prediction (D+2). The DNN model was evaluated against the currently operational Community Multiscale Air Quality (CMAQ) modelling system in South Korea. Our study demonstrated that the DNN model outperformed the CMAQ modelling results. The DNN model provided better forecasting skills by reducing the root-mean-squared error (RMSE) by 4.1 $\mu gm^{-3}$, 2.2 $\mu gm^{-3}$ and 3.0 $\mu gm^{-3}$ for the 3 consecutive days, respectively, compared to the CMAQ. Also, the false-alarm rate (FAR) also decreased by 16.9 %p (D+0), 7.5 %p (D+1), and 7.6 %p (D+2), indicating that the DNN model substantially mitigated the overprediction of the CMAQ in high $PM_{2.5}$ concentrations. These results showed that the DNN model outperformed the CMAQ model when it was simultaneously trained by using the observation and forecasting data from the numerical air quality models. Notably, the forecasting data provided more benefits to the DNN modelling results as the forecasting days increased. Our results suggest that our data-driven machine learning approach can be a useful tool for air quality forecasting when it is implemented with air quality models together by reducing model-oriented systematic biases.

**Comment.** We also found out a type such as YUS which may be YSU.

**Response.** We modified YUS to YSU.